# DNA Mismatch Repair Gene Variant Classification: Evaluating the Utility of Somatic Mutations and Mismatch Repair Deficient Colonic Crypts and Endometrial Glands

**DOI:** 10.3390/cancers15204925

**Published:** 2023-10-10

**Authors:** Romy Walker, Khalid Mahmood, Julia Como, Mark Clendenning, Jihoon E. Joo, Peter Georgeson, Sharelle Joseland, Susan G. Preston, Bernard J. Pope, James M. Chan, Rachel Austin, Jasmina Bojadzieva, Ainsley Campbell, Emma Edwards, Margaret Gleeson, Annabel Goodwin, Marion T. Harris, Emilia Ip, Judy Kirk, Julia Mansour, Helen Mar Fan, Cassandra Nichols, Nicholas Pachter, Abiramy Ragunathan, Allan Spigelman, Rachel Susman, Michael Christie, Mark A. Jenkins, Rish K. Pai, Christophe Rosty, Finlay A. Macrae, Ingrid M. Winship, Daniel D. Buchanan

**Affiliations:** 1Colorectal Oncogenomics Group, Department of Clinical Pathology, Victorian Comprehensive Cancer Centre, Melbourne Medical School, Faculty of Medicine, Dentistry and Health Sciences, The University of Melbourne, Melbourne, VIC 3000, Australia; khalid.mahmood@unimelb.edu.au (K.M.); julia.como@unimelb.edu.au (J.C.); mark.clendenning@unimelb.edu.au (M.C.); ji.joo@unimelb.edu.au (J.E.J.); peter.georgeson@unimelb.edu.au (P.G.); sharelle.joseland@unimelb.edu.au (S.J.); susan.preston@unimelb.edu.au (S.G.P.); bjpope@unimelb.edu.au (B.J.P.); daniel.buchanan@unimelb.edu.au (D.D.B.); 2University of Melbourne Centre for Cancer Research, Victorian Comprehensive Cancer Centre, Melbourne Medical School, Faculty of Medicine, Dentistry and Health Sciences, The University of Melbourne, Melbourne, VIC 3000, Australia; m.jenkins@unimelb.edu.au; 3Melbourne Bioinformatics, Melbourne Medical School, Faculty of Medicine, Dentistry and Health Sciences, The University of Melbourne, Melbourne, VIC 3052, Australia; 4Genetic Health Queensland, Royal Brisbane and Women’s Hospital, Brisbane, QLD 4006, Australia; rachel.austin@health.qld.gov.au (R.A.); helen.marfan@health.qld.gov.au (H.M.F.);; 5Clinical Genetics Unit, Austin Health, Melbourne, VIC 3084, Australia; jasmine.bojadzieva@austin.org.au (J.B.); ainsley.campbell@austin.org.au (A.C.); 6Familial Cancer Service, Westmead Hospital, Sydney, NSW 2145, Australia; emma.edwards2@health.nsw.gov.au; 7Hunter Family Cancer Service, Newcastle, NSW 2298, Australia; margaret.gleeson@health.nsw.gov.au (M.G.); judy.kirk@sydney.edu.au (J.K.); abiramy.ragunathan@health.nsw.gov.au (A.R.); 8Cancer Genetics Department, Royal Prince Alfred Hospital, Camperdown, NSW 2050, Australia; annabel.goodwin@health.nsw.gov.au (A.G.); allan.spigelman@svha.org.au (A.S.); 9Sydney Medical School, Faculty of Medicine and Health, University of Sydney, Sydney, NSW 2050, Australia; 10Monash Health Familial Cancer Centre, Clayton, VIC 3168, Australia; marion.harris@monashhealth.org; 11Cancer Genetics Service, Liverpool Hospital, Liverpool, NSW 2170, Australia; emilia.ip@health.nsw.gov.au; 12Tasmanian Clinical Genetics Service, Royal Hobart Hospital, Hobart, TAS 7000, Australia; julia.mansour@icon.team; 13Genetic Services of Western Australia, King Edward Memorial Hospital, Perth, WA 6008, Australia; cassandra.nichols@health.wa.gov.au (C.N.); nicholas.pachter@health.wa.gov.au (N.P.); 14Medical School, Faculty of Health and Medical Sciences, University of Western Australia, Perth, WA 6009, Australia; 15School of Medicine, Curtin University, Perth, WA 6102, Australia; 16St Vincent’s Cancer Genetics Unit, Sydney, NSW 2010, Australia; 17Surgical Professorial Unit, UNSW Clinical School of Clinical Medicine, Sydney, NSW 2052, Australia; 18Department of Medicine, Royal Melbourne Hospital, Melbourne Medical School, Faculty of Medicine, Dentistry and Health Sciences, The University of Melbourne, Melbourne, VIC 3052, Australia; michael.christie@mh.org.au; 19Department of Pathology, The Royal Melbourne Hospital, Melbourne, VIC 3052, Australia; 20Centre for Epidemiology and Biostatistics, School of Population and Global Health, Faculty of Medicine, Dentistry and Health Sciences, The University of Melbourne, Melbourne, VIC 3052, Australia; 21Department of Laboratory Medicine and Pathology, Mayo Clinic Arizona, Scottsdale, AZ 85259, USA; pai.rish@mayo.edu; 22Envoi Specialist Pathologists, Brisbane, QLD 4059, Australia; 23School of Biomedical Sciences, Faculty of Medicine, University of Queensland, Brisbane, QLD 4072, Australia; 24Genomic Medicine and Familial Cancer Centre, Royal Melbourne Hospital, Melbourne, VIC 3052, Australia; finlay.macrae@mh.org.au (F.A.M.); ingrid.winship@mh.org.au (I.M.W.); 25Colorectal Medicine and Genetics, The Royal Melbourne Hospital, Melbourne, VIC 3052, Australia; 26Department of Medicine, Melbourne Medical School, Faculty of Medicine, Dentistry and Health Sciences, The University of Melbourne, Melbourne, VIC 3052, Australia

**Keywords:** Lynch syndrome, DNA mismatch repair gene variant classification, DNA mismatch repair deficient crypts/glands, colorectal cancer, endometrial cancer, variant of uncertain significance, DNA mismatch repair gene somatic mutations

## Abstract

**Simple Summary:**

Lynch syndrome is caused by germline pathogenic variants in the DNA mismatch repair (MMR) genes predisposing carriers to colorectal and endometrial cancer. Genetic testing for Lynch syndrome, in the form of multigene panel testing, frequently identifies variants of uncertain clinical significance (VUS). These VUS have limited clinical actionability and create uncertainty for patients and clinicians regarding their risk of cancer. In this study, we tested carriers of germline VUS for features consistent with Lynch syndrome, namely (1) tumor microsatellite instability/MMR-deficiency, (2) the presence of a somatic second hit in the MMR gene harboring the VUS by tumor sequencing and (3) the presence of MMR-deficiency in normal colonic mucosa crypts or normal endometrial glands. Our findings showed that microsatellite instability/MMR-deficiency status and somatic second hits were consistent with MMR variant classifications as determined by the ACMG/InSiGHT framework. In addition to this, the presence of MMR-deficient crypts/glands were consistent with pathogenic variant classification.

**Abstract:**

Germline pathogenic variants in the DNA mismatch repair (MMR) genes (Lynch syndrome) predispose to colorectal (CRC) and endometrial (EC) cancer. Lynch syndrome specific tumor features were evaluated for their ability to support the ACMG/InSiGHT framework in classifying variants of uncertain clinical significance (VUS) in the MMR genes. Twenty-eight CRC or EC tumors from 25 VUS carriers (6x*MLH1*, 9x*MSH2*, 6x*MSH6*, 4x*PMS2*), underwent targeted tumor sequencing for the presence of microsatellite instability/MMR-deficiency (MSI-H/dMMR) status and identification of a somatic MMR mutation (second hit). Immunohistochemical testing for the presence of dMMR crypts/glands in normal tissue was also performed. The ACMG/InSiGHT framework reclassified 7/25 (28%) VUS to likely pathogenic (LP), three (12%) to benign/likely benign, and 15 (60%) VUS remained unchanged. For the seven re-classified LP variants comprising nine tumors, tumor sequencing confirmed MSI-H/dMMR (8/9, 88.9%) and a second hit (7/9, 77.8%). Of these LP reclassified variants where normal tissue was available, the presence of a dMMR crypt/gland was found in 2/4 (50%). Furthermore, a dMMR endometrial gland in a carrier of an *MSH2* exon 1-6 duplication provides further support for an upgrade of this VUS to LP. Our study confirmed that identifying these Lynch syndrome features can improve MMR variant classification, enabling optimal clinical care.

## 1. Introduction

Lynch syndrome is the most common hereditary cancer predisposition syndrome with an estimated carrier frequency in the population of 1 in 280 [1] and a prevalence of up to 5% in colorectal cancer (CRC) or endometrial cancer (EC) affected people [2,3,4,5]. Lynch syndrome is caused by germline pathogenic variants in one of the DNA mismatch repair (MMR) genes, *MLH1*, *MSH2*, *MSH6*, *PMS2* [6] or by deletions in the 3′ end of the *EPCAM* gene leading to transcriptional silencing of *MSH2* [7]. People with Lynch syndrome have an increased risk of not only CRC and EC but also of cancers of the ovaries, stomach, duodenum, bile duct, gall bladder, pancreas, urinary bladder, ureter, kidney, breast, prostate and brain tumors [8]. Current estimates of penetrance for CRC vary by gene and sex, but on average approximately one-third to one-half of germline MMR pathogenic variant carriers will be diagnosed with CRC by the age of 70 [9,10]. Once identified, carriers of MMR gene pathogenic variants can be offered screening via colonoscopy with polypectomy and other opportunities to prevent cancer development or the ability to diagnose Lynch syndrome cancers at an early curable stage [11].

The current diagnostic approach to identify carriers of pathogenic MMR variants involves immunohistochemical (IHC) testing of CRC or EC tumors for loss of MMR protein expression followed by germline multigene panel testing of the MMR genes [12]. A recurring outcome from genetic tests is the identification of germline variants of uncertain clinical significance (VUS) in one of the MMR genes, reported to occur in 6% of the cases undergoing testing for Lynch syndrome [13]. Uncertainty regarding the pathogenicity of a VUS impacts clinical management, as carriers of pathogenic variants receive more intensive clinical care including screening such as colonoscopy, and choice/timing of possible risk-reducing surgeries, than carriers of benign variants [14]. The American College of Medical Genetics and Genomics (ACMG) has developed standards and guidelines for the interpretation of sequence variants identified in Mendelian disorders [15]. Here, the recommendation is to classify variants into five categories (class 1–5) based on available evidence deriving from variant prevalence in the population, in silico effect prediction, functional assays and segregation data, amongst other data sources [15]. Comparably, the International Society for Gastrointestinal Hereditary Tumours (InSiGHT) working group determined InSiGHT criteria to aid with MMR variant classification (https://www.insight-group.org/criteria/, last accessed date: 31 May 2023). In this Bayesian-based analysis, variant pathogenicity probabilities derive from tumor characteristics and predetermined combinations of evidence types to predict the variant pathogenicity likelihood [16].

The tumor characteristics used in the InSiGHT criteria to guide MMR variant classification are features commonly observed in individuals diagnosed with Lynch syndrome, such as high levels of microsatellite instability (MSI-H) and MMR-deficiency (dMMR) [17], as determined by MSI polymerase chain reaction (MSI-PCR) and IHC assays, respectively. MSI-PCR and IHC each have their own limitations leading to false positive or false negative testing results [18,19], which impedes accurate classification of MMR variants. MSI-H/dMMR tumors develop in Lynch syndrome when one allele in an MMR gene becomes inactivated by a germline pathogenic variant while the other allele becomes inactivated by either a somatic mutation or by loss of heterozygosity (LOH). This acquisition of a somatic second hit, as described by the Knudson two-hit hypothesis [20], results in complete loss of MMR function. As next-generation sequencing (NGS) becomes more widely adopted for precision oncology and diagnostic purposes, the ability to accurately determine MSI-H/dMMR status [21,22] and identify MMR gene somatic mutations/LOH [23] using this methodology is becoming increasingly attractive as a streamlined approach to diagnosing Lynch syndrome. While the strength of both the ACMG guidelines and InSiGHT criteria are the ability to draw on multiple data sources, these may not always be accessible in clinical settings [16]. Therefore, despite important advances, classification of germline variants in the MMR genes remains challenging.

A novel finding in Lynch syndrome has been the identification of dMMR in morphologically normal colonic crypts [24,25,26,27] or endometrial glands [28,29]. dMMR crypts/glands are specific to people with Lynch syndrome, observed only in carriers and not in people with sporadic MSI-H/dMMR tumors [27]. In people with Lynch syndrome, the acquisition of the somatic second hit in normal tissue results in biallelic inactivation of the MMR gene, which is evidenced by loss of MMR protein expression as detected by IHC [24,27], representing the initiation of tumorigenesis. Therefore, the presence of a dMMR crypt/gland is a strong indicator of a germline MMR pathogenic variant. To date, this unique characteristic has not yet been investigated for its potential application in MMR gene variant classification approaches.

To be most inclusive of current variant classification parameters, we applied a combination of the ACMG guidelines integrated with the InSiGHT criteria, hereon referred to as the ACMG/InSiGHT framework, for classification of variants in the MMR genes. MMR gene variant classification has undergone substantial progress in the last decade through incorporation of features unique to the Lynch syndrome phenotype. With the evolution of NGS-based diagnostics, including for Lynch syndrome [30,31], the shift from tumor IHC to tumor NGS to determine MSI-H/dMMR status is gaining support. Additionally, evidence of the presence of dMMR crypts/glands has the potential to inform MMR gene variant classification within the ACMG/InSiGHT approach. Thus, in this study, we investigated the role of (1) NGS-based MSI-H/dMMR status using an additive feature combination approach as described previously [22], (2) the presence of a somatic second hit (single mutation or LOH) in the MMR gene harboring the VUS and (3) the presence of a dMMR crypt/gland in normal colonic or endometrial tissue, in classifying 25 MMR VUS and compared this to their classification status derived from the ACMG/InSiGHT framework. Determination of additional features or approaches to support MMR gene variant classification will improve the diagnosis of Lynch syndrome and the precision prevention of cancer in carriers.

## 2. Materials and Methods

### 2.1. Patient Cohort

Participants were men and women diagnosed with primary CRC or women diagnosed with EC (*n* = 25) who were identified by clinical genetic testing to carry a germline MMR gene VUS as defined by the clinical testing report and subsequently referred by one of the Family Cancer Clinics across Australia to the ANGELS study (“Applying Novel Genomic approaches to Early-onset and suspected Lynch Syndrome colorectal and endometrial cancers”) between 2018 and 2022 [32] (test group, *n* = 25 carriers who developed *n* = 28 tumors, Table 1). Cancer-affected relatives were recruited where possible to investigate segregation of the VUS.

A biopsy or resection tumor tissue specimen was collected from each participant. Normal colonic mucosa or normal endometrium tissue were collected where possible. Pedigree information was collected during clinical work-up and segregation of the MMR VUS in family members was performed via Sanger sequencing as part of this study. A group of CRC- or EC-affected MMR pathogenic variant carriers (*n* = 19) or non-carriers of a germline MMR pathogenic variant or VUS (*n* = 20) who were participants of the Australasian Colorectal Cancer Family Registry (ACCFR) were included as reference groups, where *n* = 37 underwent targeted panel sequencing or whole exome sequencing as shown in Appendix A. The molecular and phenotypic characterization of these individuals and their tumors from the ANGELS [22,32,33] and the ACCFR [2,34,35,36] have been described previously. All studies were approved by the Human Research Ethics Committees at The University of Melbourne (HREC#1750748) and hospitals governing participating family cancer clinics. Written informed consent was obtained from all participants.

### 2.2. Immunohistochemical Testing for DNA Mismatch Repair Protein Expression

For the 25 MMR VUS carriers, IHC of tumor tissue was derived from either the diagnostic pathology report or from testing performed by this study. For this study, the Ventana *DISCOVERY ULTRA* automated stainer (Ventana Medical Systems Inc., Oro Valley, AZ, USA) was used with anti-MLH1 (M1), anti-MSH2 (G219-1129), anti-MSH6 (SP93) mouse monoclonal and anti-PMS2 (A16-4) rabbit monoclonal primary antibodies (Roche Diagnostics, Basel, Switzerland) and tested on a 4 µM tissue section. All staining protocols were performed following the manufacturer’s protocol (Roche Diagnostics).

To detect dMMR colonic crypts or endometrial glands, IHC was performed on 4 µM sections from a tissue block containing non-tumor-adjacent normal colonic mucosa or endometrium from the resection margins. Twenty serial 4 µM sections were cut to a depth of 80 µM with the 1st, 10th and 19th slides stained for the MMR protein that was concordant with the MMR gene harboring the VUS. If no dMMR crypt/gland was identified, a further twenty 4 µM sections were cut and screened. This process was repeated up to three times with a maximum of 3 × 80 µM sections of tissue screened. When a dMMR crypt/gland was identified, the subsequent section (2nd, 11th or 20th slide) was stained for the unaffected MMR gene as a control for artefactual loss of expression. For example, if an MSH2-deficient crypt/gland was identified, the next slide was stained for MLH1 protein expression. All dMMR crypts/glands identified in this study were independently confirmed by two senior pathologists (CR and RP), with 100% concordance in classifying stained slides as positive or negative for dMMR crypts/glands. Six normal colonic mucosa and six normal endometrial tissue samples were available for a total of 12 VUS carriers.

### 2.3. Tumor MLH1 Methylation Testing Assays

Testing for *MLH1* gene promoter methylation on formalin-fixed paraffin embedded (FFPE) tissue DNA was performed as previously described [22,33]. Briefly, two independent *MLH1* methylation assays, namely MethyLight [2,34] and MS-HRM (methylation-sensitive high resolution melting assay) [37], were used to test the same tumor DNA sample alongside a set of DNA standards (0–100% methylation) and no-template (negative) controls. Bisulfite conversion of tumor DNA was performed using the EZ DNA Methylation-Lightning^TM^ Kit (Zymo Research, Irvine, CA, USA). For MethyLight, *MLH1* methylation was quantitatively reported based on the percentage of methylated reference (PMR) calculations [34], where tumors with a PMR ≥ 10% were considered “positive” [2,34]. For MS-HRM, the MeltDoctor^TM^ HRM Reagent Kit (Thermo Fisher Scientific, Waltham, MA, USA) was used where tumors demonstrating ≥ 5% were considered *MLH1* methylation “positive”.

### 2.4. Next-Generation Sequencing

All available FFPE tumor tissue DNA (*n* = 28) and matched blood-derived DNA samples from 25 MMR VUS carriers (Table 2) underwent targeted multigene panel sequencing using the panel capture previously described [22,33]. This customized panel incorporated 297 genes, including the MMR and *EPCAM* genes, as well as other established hereditary CRC and EC genes and the *BRAF* p.V600E mutation (2.005 megabases). Library preparation was performed using the SureSelect^TM^ Low Input Target Enrichment System from Agilent Technologies (Santa Clara, CA, USA) using standard procedures. The median on-target coverage for the panel sequenced test tumors was 906 (interquartile range = 763–1099) for the tumor DNA and 154 (interquartile range = 128–172) for blood-derived DNA samples. Panel libraries were sequenced on an Illumina NovaSeq 6000 (San Diego, CA, USA) comprising 150 base pair (bp) paired end reads performed at the Australian Genome Research Facility.

### 2.5. Bioinformatics Pipeline

Adapter sequences were trimmed from raw FASTQ files using trimmomatic (v.0.38) [38] and aligned to the GRCh37 human reference genome using Burrows-Wheeler-Aligner (v.0.7.12) to generate BAM files. Germline and somatic single nucleotide variants and somatic insertions/deletions (INDELs) were called using Strelka (v.2.9.2., Illumina, San Diego, CA, USA) and Mutect2 (v.0.5) using the recommended workflows [39,40]. Tumor mutational signatures (TMS) were calculated using the pre-defined set of 18 small (1–50 bp) insertions/deletions (ID) signatures as published on COSMIC (https://cancer.sanger.ac.uk/signatures/, last accessed date: 13 January 2023, v.3.2) [41]. All variants were restricted to the panel capture region and filtered based on PASS variants called by both Strelka and Mutect2. All variants were further filtered on a minimum depth of 50bp for normal/blood and tumor samples with a minimum variant allele frequency of 10% [32]. The tumor mutation burden was calculated as the number of somatic single nucleotide and INDEL mutations divided by the target region (megabase). For variant calling, the following RefSeq transcripts were used (*MLH1*: NM_000249.3, *MSH2*: NM_000251.2, *MSH6*: NM_000179.2 and *PMS2*: NM_000535.5). The MMR genes were interrogated for somatic mutations, including single somatic mutations (e.g., missense, nonsense, insertion, deletion, frameshift variant type) and LOH, in the same gene as the germline MMR VUS. LOH across the MMR genes were called using LOHdeTerminator (v.0.6, https://github.com/supernifty/LOHdeTerminator, accessed on 6 October 2023). The pathogenicity of somatic MMR mutations were determined using the Varsome database [42] (https://varsome.com/, last accessed date: 13 January 2023), which categorizes variants into the ACMG classification system. All likely pathogenic/pathogenic MMR mutations were manually confirmed in BAM files using the Integrative Genomics Viewer (v.2.3) [43].

### 2.6. Determination of Tumor Microsatellite Instability and Mismatch Repair Deficiency

Panel-sequenced tumors were assessed for evidence of MSI-H/dMMR using: (1) four independent MSI detection tools, namely MSMuTect [44], MANTIS [45], MSIseq [46] and MSISensor [47], (2) INDEL count and (3) the combination of ID2 TMS with ID7 TMS (TMS ID2 + ID7) [32] as described in Walker et al. 2023 [22]. Overall tumor MSI-H/dMMR status was determined by combining these six features (using an additive feature combination approach), where a tumor with any 3 or more of these 6 features with positivity for dMMR was considered to be MSI-H/dMMR [22]. This approach has been shown to be the most robust across whole-exome sequencing and panel assays as well as across CRC and EC tumors, while presenting with the highest prediction accuracy to differentiate dMMR from pMMR (MMR-proficient) tumors [22].

### 2.7. Classifying MMR Variants Using a Combination of Existing Methodologies

We used a combination of the ACMG [15] and InSiGHT [16] criteria, hereon referred to as the ACMG/InSiGHT framework, for improved and contemporary variant classification. Briefly, the features assessed are displayed in Table 2, including:1.Rarity of MMR variant (the rarer a variant, the more likely the variant will not be present in healthy controls, with <1 in 50,000 alleles indicating MMR variant rarity in gnomAD using the non-cancer dataset);2.Incorporation of tumor characteristics generated by this study, including age of diagnosis, tumor NGS-derived MSI-H/dMMR status, tumor *BRAF* V600E mutation status, tumor *MLH1* methylation status and MMR IHC result;3.Prior probability scores calculated for missense variants using the in silico prediction tools Multi-variate Analysis of Protein Polymorphisms [48] and PolyPhen-2.1 [49] (pre-computed prior probabilities with a score of >0.68 and ≤0.81 indicate variant pathogenicity as determined in https://hci-priors.hci.utah.edu/PRIORS/ (last accessed date: 18 July 2023));4.Tumor characteristics, either generated from this study or available from external public data, for the same variant were combined to generate a tumor odds pathogenicity score [50,51];5.Evidence of functional effect on protein structure (e.g., ClinVar, https://www.ncbi.nlm.nih.gov/clinvar/, last accessed date: 1 June 2023);6.Co-segregation of variant with disease phenotype with a combined Bayes Likelihood Ratio >18.7 in two or more families [16] (e.g., COsegregation v.2: https://fengbj-laboratory.org/cool2/manual.html, last accessed date: 1 June 2023);7.Predicted splicing effect using SpliceAI (with a delta score of >0.2 indicating pathogenicity) (https://spliceailookup.broadinstitute.org/, last accessed date: 1 June 2023) [52].

These parameters were cumulatively considered for final MMR variant classification and variants were categorized into the recommended five-tier ACMG classifications (class 5—pathogenic (P), class 4—likely pathogenic (LP), class 3—variant of uncertain significance (VUS), class 2—likely benign (LB), class 1—benign (B)) [15]. The final ACMG/InSiGHT classification for each of the 25 MMR VUS was then assessed for concordance with the tumor NGS-derived MSI-H/dMMR status, somatic second hit and dMMR crypt/gland testing results.

## 3. Results

### 3.1. Characteristics of Patients with MMR VUS

An overview of the study design is shown in Figure 1. A total of 24 carrier families with 25 unique germline MMR VUS were included in the study comprising VUS in *MLH1* (*n* = 6), *MSH2* (*n* = 9), *MSH6* (*n* = 6) and *PMS2* (*n* = 4) with 14/25 (56%) VUS resulting in missense changes (Table 1 and Table 2). Testing for segregation of the VUS in relatives identified an additional 18 carriers, where the cancer-affected status of each of the 42 germline VUS carriers (probands and relatives) included in the study are shown in Table 1. The tumor type and age at diagnosis (ranging from 19–82 years) for each of the carriers are listed in Table 1, where 16/42 (38.1%) carriers developed multiple tumors. The pedigree for each of the VUS carrying families is provided in Appendix A. For 25/42 (59.5%) VUS carriers in this study, we tested one or more tumors (Table 1). The pattern of loss of MMR protein expression by IHC was concordant with the MMR gene harboring the VUS in 13/25 (52%) of the cases, discordant in 8/25 (32%) of the cases, while for a further four carriers (16%), no loss of MMR protein expression by IHC was reported (pMMR) (Table 1 and Table 2).

### 3.2. Variant Classification Using the ACMG/InSiGHT Framework

The re-classification of the 25 MMR VUS based on the ACMG/InSiGHT framework (see Methods above) is shown in Table 2. A total of ten out of twenty-five (40%) VUS were reclassified, with seven VUS (28%) reclassified as likely pathogenic (class 4), all of which showed a concordant pattern of MMR protein loss of expression by IHC. Two out of twenty-five (8%) were reclassified as likely benign (class 2) and one VUS was reclassified as benign (class 1), where each of the three LB/B variants had a pattern of MMR protein loss that was discordant with the MMR gene harboring the variant (Table 2). None of the VUS were categorized as pathogenic (class 5). For the remaining 15 VUS, the additional information provided by this study did not change their classification as a class 3 variant. Of these, six VUS were concordant with the observed IHC pattern of loss, five VUS were discordant to the observed IHC pattern of loss, with four VUS displaying no MMR protein loss by IHC (pMMR) (Table 2).

### 3.3. Determining Microsatellite Instability/DNA Mismatch Repair Deficiency Using Tumor Sequencing Data

To assess whether tumor features and somatic profiles generated using tumor panel sequencing could inform MMR VUS classification, tumor features associated with Lynch syndrome were assessed on 28 tumors collected from the 25 VUS carriers in this study (Table 1), and compared with a reference group of dMMR tumors from known germline MMR pathogenic variant carriers (*n* = 16) and pMMR tumors from non-MMR carriers (*n* = 18) that previously underwent tumor panel sequencing (Appendix A). The aim was to determine tumor MSI/MMR status from NGS by applying the previously described additive feature combination approach [22] and compare this with the observed MMR IHC status. For the reference group of tumors, the MSI/MMR status from tumor sequencing was 100% concordant with the MMR IHC dMMR or pMMR result. As expected for the Lynch syndrome tumors, the pattern of MMR protein loss was concordant with the MMR gene harboring the germline pathogenic variant and all were MSI-H/dMMR by tumor sequencing (Appendix A, Appendix A). In the test group of 28 tumors from 25 VUS carriers, 25/28 were classified as MSI-H/dMMR from tumor sequencing, of which 23/25 (92%) were dMMR by MMR IHC (Table 3, Appendix A). Three tumors were classified as MSS/pMMR from tumor sequencing, with two (66%) of these also confirmed to be pMMR by MMR IHC (Table 3, Appendix A). All seven of the VUS reclassified by ACMG/InSiGHT framework to LP had at least one tumor that was MSI-H/dMMR (Table 3). There were an additional eight tumors from seven VUS carriers that were not reclassified by ACMG/InSiGHT framework that were MSI-H/dMMR from tumor sequencing and demonstrated loss of the MMR protein/s concordant with the MMR gene harboring the VUS (Table 3).

### 3.4. Determining Somatic MMR Gene Second Hit Using Tumor Sequencing Data

Following the “two-hit” model for Lynch syndrome, a somatic second hit would be expected in the same gene as the gene carrying the germline variant. We aimed to identify the presence of a somatic MMR mutation as the second hit in the gene carrying the germline MMR VUS. For the reference group of dMMR Lynch syndrome tumors, 13 of the 16 (81.3%) harbored a detectable somatic second hit in the same MMR gene harboring the germline pathogenic variant, which included single somatic mutations (6/13, 46%) and LOH (7/13, 53.8%) (Appendix A, Appendix A). Of the 18 reference pMMR non-MMR carrier tumors, only 2/18 (11.1%) harbored a somatic mutation in one of the four MMR genes, both of which were LOH events (Appendix A, Appendix A). The findings from these reference tumors highlights the enrichment of somatic MMR mutations as second hits in Lynch-syndrome-related CRCs and ECs. Similarly, in the test group, for 86.7% of the cases (13/15), a second hit could be identified where the MMR VUS was concordant to the IHC pattern of loss. The second hit was more commonly a single somatic mutation (Appendix A). Equivalent to the reference group, for cases with no MMR loss of protein expression by IHC, the second hit was exclusively of the LOH mutation type (Appendix A).

For seven out of nine (77.8%) tumors from the seven VUS reclassified by ACMG/InSiGHT framework to LP, a second hit was identified (Table 3). There were two exceptions. The first was an EC diagnosed at 57 years from person ID_315-2 who carried the *MSH2* c.1862G > T p.(Arg621Leu) variant, but the tumor showed loss of MLH1/PMS2^+^ expression related to tumor *MLH1* promoter methylation. The sister (ID_315) was also a carrier and developed an EC at 62 years, which demonstrated loss of MSH2/MSH6 expression and a somatic second hit in *MSH2.* The second exception was an EC diagnosed at 53 years from ID_156 who carried the *MSH6* c.3556 + 5_3556 + 9delins variant that was MSI-H/dMMR from tumor sequencing, showed solitary loss of MSH6 expression but no somatic mutation in *MSH6*; however, a CRC diagnosed at 61 years in ID_156 also showed a solitary loss of MSH6 expression with a second hit in *MSH6.*

There were an additional eight tumors from seven VUS carriers that were not reclassified by ACMG/InSiGHT framework, but demonstrated a second hit (Table 3), including ID_176 who carried the *MLH1* c.1594G > C p.(Gly532Arg) variant and developed MLH1/PMS2 deficient CRC and EC, in the absence of *MLH1* methylation, where a second hit was observed in both tumors. Two VUS carriers, ID_170 (*MLH1* c.-117G > T p.?) and ID_111 (*MSH6* c.1153_1155del p.(Arg385del)), demonstrated MSI-H/dMMR by tumor sequencing and a second hit in their respective CRCs; however, both CRCs were pMMR by IHC, suggesting a false negative MMR IHC result. An additional VUS carrier, ID_376 (*MSH2* exon 1-6 duplication) whose CRC showed loss of *MSH2/MSH6* by IHC, MSI-H/dMMR by tumor sequencing, did not harbor a second hit in the *MSH2* gene (Table 3).

There were a further eight VUS carriers whose tumors were all MSI-H/dMMR by tumor sequencing, but their pattern of MMR protein loss by IHC indicated a different MMR gene was defective to the one harboring the VUS (Table 3). Tumor sequencing revealed two somatic MMR mutations (also known as “double somatics”), which were likely responsible for the pattern of MMR protein loss by IHC in these eight carriers. For two of these VUS (ID_263: *MSH6* c.*85T > A and ID_202: *MSH2* c.138C > G p.(His46Gln)), a second hit was observed; however, the ACMG/InSiGHT framework reclassified these variants as benign and likely benign, respectively (Table 3).

The specificity of the somatic MMR mutations to the MMR gene harboring the VUS was assessed. For the reference group of dMMR Lynch syndrome tumors, a somatic second hit was identified in 100% of *MLH1* (*n* = 4), in 70% of *MSH2* (*n* = 10), in 100% of *MSH6* (*n* = 1) and 100% of *PMS2* (*n* = 1) germline pathogenic variant carriers, but somatic MMR mutations in the other MMR genes were rarely observed (Appendix A). For the reference pMMR non-Lynch syndrome tumors, the presence of any MMR somatic mutation was found in only 11.1% (2/18) of the cases screened (Appendix A). In the test group, for the MMR VUS categorized as LP by the ACMG/InSiGHT framework, only three out of nine (33.3%) tumors presented with ≥1 somatic event in the gene that did not harbor the germline VUS (Table 3, Figure 2). For the three cases where the germline VUS was classified as LB/B, a different molecular mechanism, e.g., double somatic MMR mutations (*n* = 2) or a concomitant germline variant plus somatic second hit in the same gene (*n* = 1, ID_161), is the likely cause for the observed tumor dMMR (Table 3, Figure 2).

### 3.5. Detection of DNA Mismatch Repair Deficient Crypts/Glands in Normal Tissue

To establish the protocol, screening of normal colonic mucosa for dMMR crypts was performed for three pathogenic variant carriers from the reference group with available tissue. Two crypts from two different carriers demonstrated a loss of expression of the MLH1 protein, which was concordant with the germline pathogenic variant in *MLH1* (Figure 3A, Appendix A and Table 4). The screening did not identify a dMMR crypt in the third reference case (Ref_411) from 2 × 80 µM tissue screening, after which the tissue was depleted. In addition to finding a dMMR crypt in the normal colonic mucosa of *MLH1* pathogenic variant carrier Ref_029, a dMMR crypt was identified in the normal colonic mucosa of their maternal aunt (Ref_029-2), who was also a carrier of the family *MLH1* pathogenic variant (Table 4). No dMMR crypts were identified from 3 × 80 µM tissue screening of the normal colonic mucosa from two CRC-affected people who did not carry a germline MMR pathogenic variant (Table 4).

For 12/25 (48%) of MMR VUS carriers, normal tissue specimens (6× normal colonic and 6× normal endometrial tissue) were available for dMMR crypt/gland screening. A single case (ID_143) was a technical failure for PMS2 IHC staining due to poor tissue fixation of the normal tissue. In 7/11 (63.6%) of the remaining cases, normal tissue blocks were available for screening while for 4/11 (36.4%) of the cases, only 4 µM normal tissue sections on slides were available for screening. A total of three carriers (27.3%) had a dMMR crypt/gland identified out of eleven carriers tested (Table 4, Figure 3B). Two of these were in normal endometrium tissue with the remaining dMMR crypt identified in normal colonic mucosa. Out of four cases that were reclassified as likely pathogenic based on the ACMG/InSiGHT criteria and where normal tissue was available for testing, two (ID_315: *MSH2* c.1862G > T p.(Arg621Leu) and ID_156: *MSH6* c.3556 + 5_3556 + 8delins) had a dMMR gland and dMMR crypt identified, respectively (Table 4). A dMMR endometrial gland was identified in the carrier of *MSH2* exon 1-6 duplication (ID_058), where the tumor also demonstrated MSI-H/dMMR by tumor sequencing, a somatic second hit in *MSH2* and showed loss of MSH2/MSH6 expression by IHC; however, the ACMG/InSiGHT framework did not result in a reclassification of the VUS (Table 4). The pathogenic criterion (PVS1) from the ACMG guidelines could not be applied for a predicted loss of function as the location of the partial gene duplication was unknown, making it uncertain if nonsense-mediated mRNA decay would take place [53].

## 4. Discussion

In this study, tumor and non-malignant tissue features associated with germline pathogenic MMR variant carriers were investigated to determine their utility to aid MMR variant classification. Our findings from the investigation of 28 tumors from 25 VUS carriers showed that tumor MSI-H/dMMR status, determined by tumor sequencing and an additive feature combination approach [22], agreed with variant LP/P classification (Figure 4). We found MSI-H/dMMR status by tumor sequencing was 100% concordant with dMMR status by IHC in both our reference group of dMMR Lynch syndrome tumors (Appendix A) and in the tumors from seven VUS carriers that were reclassified to LP by the ACMG/InSiGHT framework (Table 3), while the reference group of pMMR non-MMR carrier tumors were MSS/pMMR by tumor sequencing (Appendix A). Furthermore, the identification of a somatic second hit was also consistent with variant LP/P classification. A second hit was observed in 81.3% of the reference group of dMMR Lynch syndrome tumors (Appendix A) and 77.8% of the tumors from VUS reclassified to LP (Table 3) in contrast to only 11.1% of tumors from the reference group of pMMR non-MMR carriers having a somatic MMR mutation (Appendix A). In light of these findings, a further seven VUS, that could not be reclassified by the ACMG/InSiGHT framework demonstrated tumors with MSI-H/dMMR and a second hit, suggesting that these seven VUS could be upgraded to an LP classification (Table 3). Screening for the presence of a dMMR crypt/gland also showed potential for clinical utility for LP/P variant classification. In addition to the three known pathogenic variant carriers from the reference group, three additional VUS carriers were found to have a dMMR crypt/gland, where in two of these the ACMG/InSiGHT framework reclassified the VUS to LP (Table 4). The remaining VUS case with a dMMR endometrial gland was identified in the carrier of *MSH2* exon 1-6 duplication (ID_058), and together with the tumor also demonstrating MSI-H/dMMR by tumor sequencing and a somatic second hit, is supportive of an LP classification for this variant (Table 4). Therefore, the application of tumor sequencing for MSI/dMMR status and the presence of a second hit together with testing for dMMR crypts/glands is likely to improve MMR variant classification.

A recent study has investigated the benefit of identifying the somatic second hit for variant classification. Scott et al. (2022) showed that somatic second hit mutations in *MSH2* were significantly more common in tumors from *MSH2* missense variant carriers that had multiplexed analysis of variant effect (MAVE) data, indicating the germline variant was functionally disruptive (i.e., pathogenic variant) when compared with tumors from *MSH2* missense variant carriers with MAVE scores indicating the germline variant was functionally normal (i.e., benign variant) [53]. This supports the observations from this study where a somatic second hit was more prevalent in both known pathogenic variant carriers as well as VUS that were reclassified to LP but rare in pMMR tumors.

A study performed by Shirts et al. (2018) demonstrated that tumor mutations in the MMR genes can support both pathogenic and benign variant classification by identifying somatic driver mutations compared with passenger mutations in patients with unexplained dMMR (i.e., suspected Lynch syndrome or Lynch-like syndrome) [54]. Furthermore, the authors propose that the cumulative evidence from independent mutations identified from sequencing unexplained dMMR tumors will ultimately classify more germline MMR gene variants. Given the rarity of some individual constitutional MMR gene variants, the observation of these same variants as somatic mutations in multiple dMMR tumors may expedite their classification. The detection of a somatic second hit, as we have shown in this study, as well as the work described by Shirts and colleagues, demonstrates that the detection of somatic MMR mutations in tumors, with confirmed MSI-H/dMMR status, can support MMR variant classification and warrants modifications of the ACMG/InSiGHT MMR variant classification guidelines to incorporate the characteristics of somatic mutations from tumor sequencing data.

An important finding from this study was the identification of double somatic MMR mutations in an MMR gene that was not the gene harboring the VUS. Double somatic MMR mutations are a recognized cause of somatic biallelic MMR gene inactivation that can lead to a tumor MSI-H/dMMR phenotype [23,33,55,56]. The additional information provided by the pattern of MMR protein loss by IHC was supportive that the MSI-H/dMMR tumor phenotype was caused by two somatic MMR mutations and not related to the VUS. Two of these VUS were reclassified as LB/B, supporting the somatic mutation data but in MSI-H-dMMR tumors (Table 3). A caveat to these findings was the presence of two somatic *MSH2* mutations in the CRC from person ID_138 carrying the *MSH2* c.328A > C p.(Lys110Gln) VUS (Table 3). One of these two somatic *MSH2* mutations may represent the second hit to the germline VUS; however, the two somatic *MSH2* mutations may represent somatic biallelic inactivation (Table 3). Of interest, the MAVE data for this *MSH2* missense VUS suggest it is likely benign [57] supporting a “double somatic” rather than a germline cause of MSI-H/dMMR for this tumor. Consideration of the number of somatic MMR mutations identified together with MMR IHC findings will help to interpret tumor sequencing data for MMR variant classification.

There were four tumors from four carriers where no loss of MMR protein expression was observed by IHC (Table 3). Two tumors (CRC_240 and CRC1_352)) were MSS/pMMR by tumor sequencing supporting IHC result. The other two tumors (CRC_170 and CRC_111) were MSI-H/dMMR by NGS and showed a somatic second hit in the gene with the VUS (both LOH events), which may suggest a false negative MMR IHC result.

The presence of a dMMR crypt or gland is a strong predictor for a variant being pathogenic given its specificity for Lynch syndrome [24,25,26,27,28,29]. In this study, a single endometrial gland showed loss of MSH2 expression in the patient harboring the *MSH2* exon 1-6 duplication (ID_058), which would support this variant being pathogenic (Table 4). The absence of detectable dMMR crypts/glands does not conversely support a LB/B classification and could simply reflect insufficient tissue was screened. The exact prevalence of dMMR crypts/glands across normal tissues still needs to be assessed in ancillary studies; however, Kloor et al. (2012) have indicated the detection of dMMR crypts in 1 cm^2^ of colonic mucosa in Lynch syndrome patients [24]. The feasibility in terms of the amount of biopsy needed to obtain at least 1 cm^2^ and cost-effectiveness of screening for dMMR crypts/glands in clinical setting needs to be determined, but may offer an alternate approach to reclassify an MMR variant, particularly when evidence from the existing ACMG/InSiGHT framework is insufficient.

A strength of this study was the comparison of data from the existing gold standard MMR variant classification framework to the application of novel features, particularly those derived from NGS, which is increasing in clinical diagnostics. The detection of MSI-H/dMMR and a second hit from tumor sequencing is unlikely to be influenced by the type of variant. Further studies are needed to determine if the detection of dMMR crypts/glands is likely to be influenced by variant type. Furthermore, the implementation of screening for dMMR crypts or glands would be based on established MMR protein antibodies and immunohistochemical protocols and, therefore, potentially more applicable to a broader spectrum of laboratories once the tissue is available. A further demonstrated strength by this study was the ability to detect dMMR crypts/glands on FFPE archival tissue that was up to 20 years old (e.g., reference group tissue) with only a single case (ID_143) failing testing. 

This study has several limitations. An important caveat for interpreting the presence of a dMMR crypt/gland for VUS classification is the concept that another undetected pathogenic variant underlies the dMMR crypt/gland rather than the VUS. Therefore, interpretation of the presence of a dMMR crypt/gland should not be considered on its own but together with additional information used to classify MMR variants. Another limitation of the study was access to normal tissue as we were only able to acquire a normal tissue specimen for half of the cases (48%, 12/25). Broader recognition that screening for dMMR crypts/glands has utility for variant classification may encourage better collection and access to normal tissue. A single Lynch syndrome tumor phenocopy was identified in the case of ID_315-2, where the tumor was positive for *MLH1* methylation despite the person carrying the *MSH2* LP variant. Although phenocopies in Lynch syndrome are rare, the interpretation of tumor data for MMR variant classification needs detailed examination. Lastly, it is possible that somatic second hits were missed in some of the tumors. This was evident for the Lynch syndrome reference tumors where a second hit was identified in only 81.3% of the sequenced tumors. Challenges in identifying more complex/cryptic variants from capture-based sequencing data or the possibility the second hit is an intronic variant not targeted by the capture may explain the missing second hits. These challenges may underlie second hit detection in the VUS cases tested in this study where, for example, the EC from person ID_156 who carried the *MSH6* c.3556 + 5_3556 + 9delins variant and CRC from person ID_376 who carried the *MSH2* exon 1-6 duplication did not identify a second hit despite the other cumulative evidence suggesting these variants are likely pathogenic. Lastly, complementary data could be gained from functional assays such as RT-PCR or minigene constructs to provide further functional evidence to support variant classification.

## 5. Conclusions

This study evaluated novel approaches to classify MMR variants, providing support for their potential incorporation into current variant classification guidelines as additional independent lines of evidence to aid MMR variant classification. Currently, somatic MMR mutation data are not used in MMR gene variant classification frameworks, but this study and other studies provide support for information gained from sequencing of dMMR tumors. Although the presence of a somatic second hit was concordant with LP/P variant classification, the knowledge that the presence of two somatic MMR gene mutations (double somatics) can also result in MSI-H/dMMR tumor phenotype needs to be acknowledged when interpreting tumor sequencing findings for variant classification. Furthermore, somatic MMR mutation data from tumor sequencing need to be considered in conjunction with confirmation the tumor is MSI-H/dMMR. The presence of a dMMR crypt/gland in normal colonic or endometrial tissue represents a novel approach to guide LP/P MMR variant classification. The identification of germline MMR VUS prior to surgery may facilitate the preservation of more normal tissue for testing, but the application of dMMR crypt/gland detection using normal colonic biopsies from colonoscopy in unaffected VUS carriers needs further investigation. Our findings have shown the potential utility of tumor sequencing to determine both MSI/MMR status and presence of single point mutation/LOH as a somatic second hit, and with assessment of normal tissue for the presence of dMMR crypts/glands for improving MMR variant classification and warrants consideration for inclusion in the ACMG/InSiGHT framework.

## Figures and Tables

**Figure 1 cancers-15-04925-f001:**
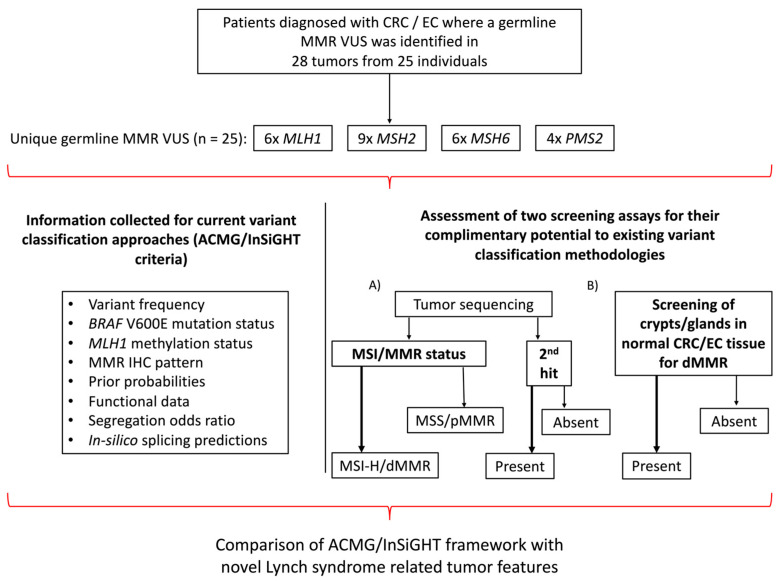
Overview of study design. Schema presenting the study inclusion criteria, the breakdown of the germline MMR VUS distribution and the testing assays applied. Abbreviations: CRC, colorectal cancer; EC, endometrial cancer; MMR, DNA mismatch repair; VUS, variant of uncertain clinical significance; IHC, immunohistochemistry; MSI, microsatellite instability; MSI-H, high levels of microsatellite instability; MSS, microsatellite stable; dMMR, DNA mismatch repair deficiency; pMMR, DNA mismatch repair proficiency; ACMG, American College of Medical Genetics and Genomics; InSiGHT, International Society for Gastrointestinal Hereditary Tumours.

**Figure 2 cancers-15-04925-f002:**
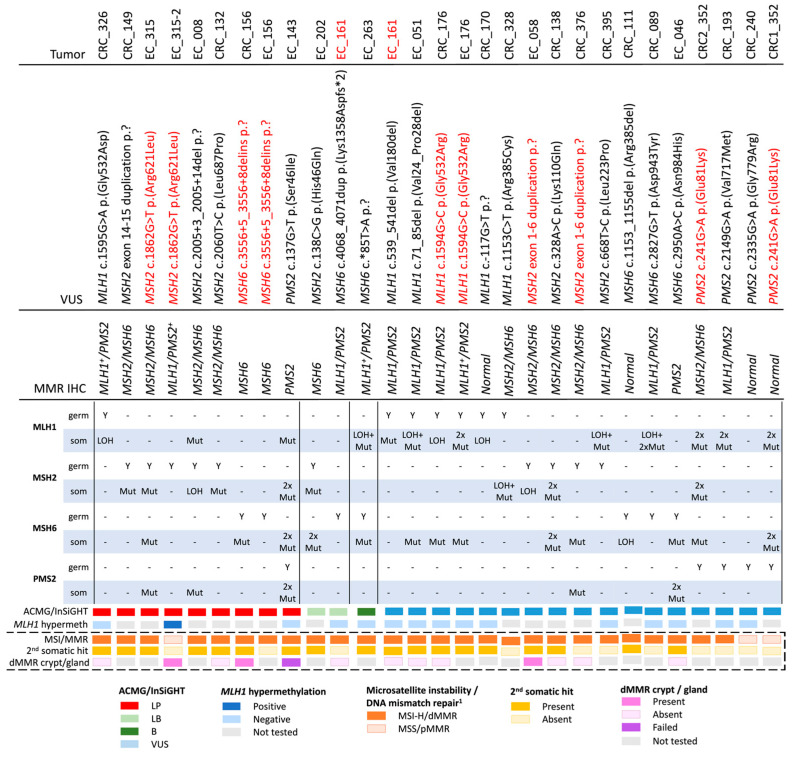
Overview of the number and type of somatic events by the 25 germline DNA mismatch repair variants of uncertain significance screened in this study. Red highlighted text indicates duplicate entries per row. Abbreviations: VUS, variant of uncertain significance; germ, germline variant; som, somatic mutation; ACMG, American College of Medical Genetics and Genomics; InSiGHT, International Society for Gastrointestinal Hereditary Tumours; MMR, DNA mismatch repair; dMMR, DNA mismatch repair deficient; pMMR, DNA mismatch repair proficient; IHC, immunohistochemistry; MSI, microsatellite instability; MSI-H, high levels of microsatellite instability; MSS, microsatellite stable; LOH, loss of heterozygosity; LP, likely pathogenic; LB, likely benign; B, benign. + Indicates heterogeneous/patchy loss of DNA mismatch repair protein expression by IHC. ^1^ Determination of the MSI/MMR status using the additive feature combination approach as previously described in Walker et al. 2023 [22].

**Figure 3 cancers-15-04925-f003:**
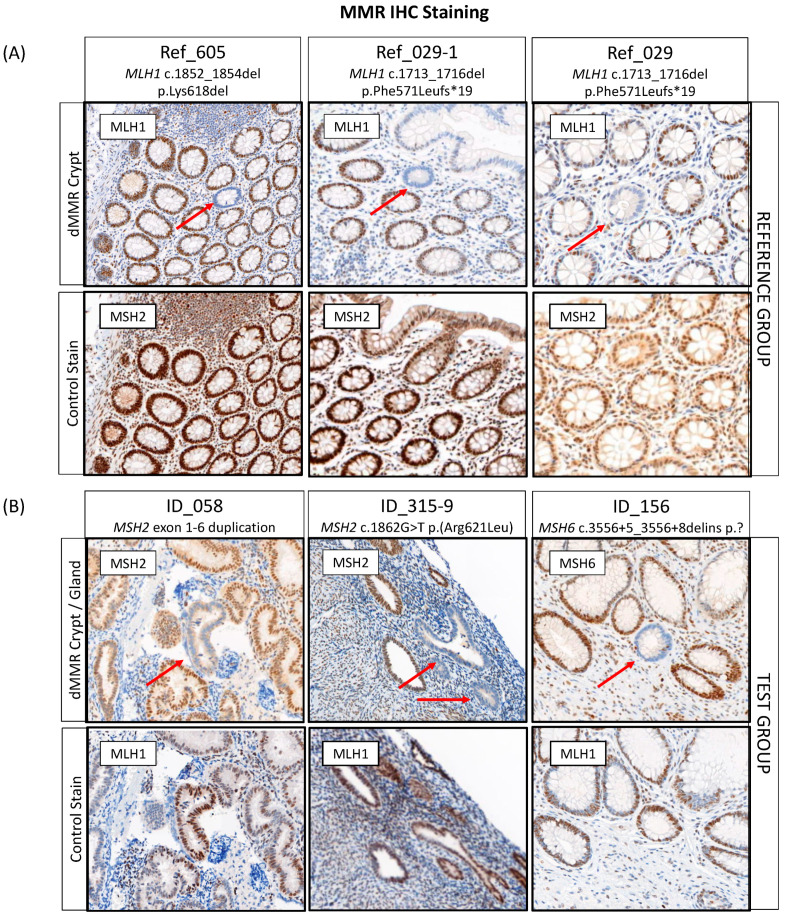
Detection of DNA mismatch repair deficient (dMMR) crypts or glands in (**A**) the reference and (**B**) the test group with three cases identified each. Red arrows indicate the location of a dMMR crypt or gland. Abbreviations: MMR, DNA mismatch repair; dMMR, DNA mismatch repair deficient; IHC, immunohistochemistry. * Indicates the variant changes the amino acid to a stop codon.

**Figure 4 cancers-15-04925-f004:**
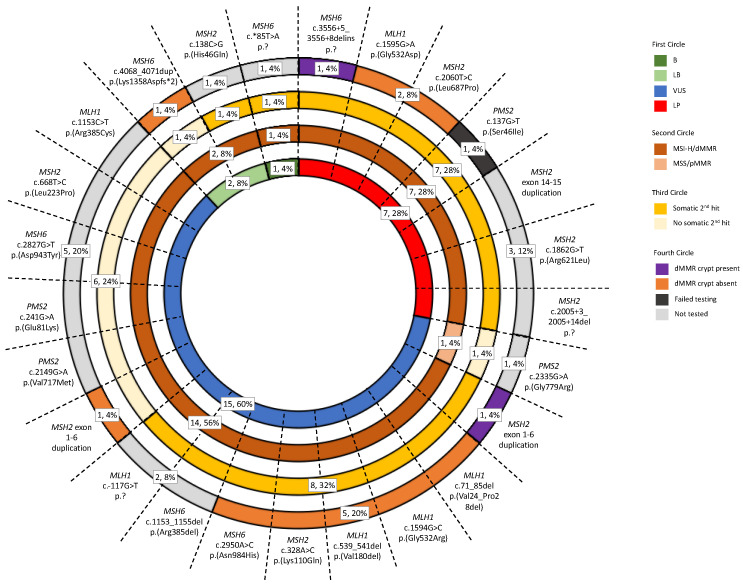
Sunburst diagram displaying the prevalence of the three Lynch-syndrome-associated features in the test group. The diagram incorporates the tumor sequencing and dMMR crypt/gland screening for determining pathogenicity against the ACMG/InSiGHT framework for MMR VUS included in the test group. Abbreviations: MMR, DNA mismatch repair; dMMR, DNA mismatch repair deficient; pMMR, DNA mismatch repair proficient; MSI, microsatellite instability; MSI-H, high levels of microsatellite instability; MSS, microsatellite stable; VUS, variant of uncertain clinical significance; LP, likely pathogenic; LB, likely benign; B, benign. * The symbol indicates the variant is located in the 3’ untranslated region in the case of *MSH6* c.*85T>A and indicates the variant changes the amino acid to a stop codon in the case of *MSH6* c.4068_4071dup p.(Lys1358Aspfs*2).

**Table 1 cancers-15-04925-t001:** List of carriers of a germline DNA mismatch repair variant of uncertain clinical significance and their cancer-affected status that were included in this study.

	Family Count	Family ID	Relationship	Carrier Count	Carrier ID	Sex	VUS Count	Gene	Variant	Variant Type	Carrier Status	Tumors Sequenced	Tissue ID	Tissue	Age at Diagnosis	IHC	*MLH1* Methylation	Carrier Tumors Tested in This Study
PROBANDS’ MMR VUS CONCORDANT TO IHC PATTERN OF LOSS	1	F_051	Index	**1**	ID_051	F	1	*MLH1*	c.71_85del p.(Val24_Pro28del)	Inframe deletion	Proband—tumor 1	1	EC_051	EC	59	MLH1/PMS2	NEGATIVE	1
		F_051	Index		ID_051	F		*MLH1*	c.71_85del p.(Val24_Pro28del)	Inframe deletion	Proband—tumor 2		Breast_051	Breast	46	NT	NT	
		F_051	Index		ID_051	F		*MLH1*	c.71_85del p.(Val24_Pro28del)	Inframe deletion	Proband—tumor 3		Duo_051	Duodenal	58	NT	NT	
		F_051	Index		ID_051	F		*MLH1*	c.71_85del p.(Val24_Pro28del)	Inframe deletion	Proband—polyp 1		Polyp1_051	Tubulovillous adenoma	55	Normal	NT	
		F_051	Index		ID_051	F		*MLH1*	c.71_85del p.(Val24_Pro28del)	Inframe deletion	Proband—polyp 2		Polyp2_051	Tubulovillous adenoma	56	Normal	NT	
		F_051	Index		ID_051	F		*MLH1*	c.71_85del p.(Val24_Pro28del)	Inframe deletion	Proband—polyp 3		Polyp3_051	Tubulovillous adenoma	59	MLH1 ^+^/PMS2	NT	
	2	F_161	Index	**2**	ID_161	F	2	*MLH1*	c.539_541del p.(Val180del)	Inframe deletion	Proband—tumor 1	2	EC_161	EC	59	MLH1/PMS2	NEGATIVE	2
							3	*MSH6*	c.4068_4071dup p.(Lys1358Aspfs*2)	Frameshift								
		F_161	Index		ID_161	F		*MLH1*	c.539_541del p.(Val180del)	Inframe deletion	Proband—tumor 2		CRC_161	CRC	34	Normal	NT	
		F_161	Son	**3**	ID_161-2 ^1^	M		*MLH1*	c.539_541del p.(Val180del)	Inframe deletion	Carrier		CRC_161-2	CRC	35	Normal	NT	
		F_161	Sister	**4**	ID_161-3 ^1^	F		*MLH1*	c.539_541del p.(Val180del)	Inframe deletion	Carrier—tumor 1		CRC1_161-3	CRC	36	NT	NT	
		F_161	Sister		ID_161-3 ^1^	F		*MLH1*	c.539_541del p.(Val180del)	Inframe deletion	Carrier—tumor 2		CRC2_161-3	CRC	61	MLH1/PMS2	NT	
		F_161	Sister		ID_161-3 ^1^	F		*MLH1*	c.539_541del p.(Val180del)	Inframe deletion	Carrier—tumor 3		Lung_161-3	Lung	63	NT	NT	
	3	F_176	Index	**5**	ID_176	F	4	*MLH1*	c.1594G > C p.(Gly532Arg)	Missense	Proband—tumor 1	3	CRC_176	CRC	47	MLH1/PMS2	NEGATIVE	3
		F_176	Index		ID_176	F		*MLH1*	c.1594G > C p.(Gly532Arg)	Missense	Proband—tumor 2	4	EC_176	EC	54	MLH1 ^+^/PMS2	NEGATIVE	
		F_176	Mother	**6**	ID_176-2	F		*MLH1*	c.1594G > C p.(Gly532Arg)	Missense	Carrier		CRC_176-2	CRC	82	PMS2	NT	
	4	F_326	Index	**7**	ID_326	F	5	*MLH1*	c.1595G > A p.(Gly532Asp)	Missense	Proband	5	CRC_326	CRC	41	MLH1 ^+^/PMS2	NEGATIVE	4
	5	F_376	Index	**8**	ID_376 ^2^	M	6	*MSH2*	Exon 1-6 duplication	Exon duplication	Proband	6	CRC_376	CRC	29	MSH2/MSH6	NT	5
	6	F_058	Index	**9**	ID_058 ^3^	F	7	*MSH2*	Exon 1-6 duplication	Exon duplication	Proband—tumor 1	7	EC_058	EC	49	MSH2/MSH6	NT	6
		F_058	Index		ID_058	F		*MSH2*	Exon 1-6 duplication	Exon duplication	Proband—tumor 2		Skin_058	Skin	47	NT	NT	
		F_058	Daughter	**10**	ID_058-2	F		*MSH2*	Exon 1-6 duplication	Exon duplication	Carrier		Polyp_058-2	Sessile serrated adenoma	19	NT	NT	
		F_058	Mother	**11**	ID_058-3	F		*MSH2*	Exon 1-6 duplication	Exon duplication	Carrier—tumor 1		CRC1_058-4	CRC	52	NT	NT	
		F_058	Mother		ID_058-3	F		*MSH2*	Exon 1-6 duplication	Exon duplication	Carrier—tumor 2		CRC2_058-4	CRC	54	NT	NT	
		F_058	Maternal uncle	**12**	ID_058-4	M		*MSH2*	Exon 1-6 duplication	Exon duplication	Obligate carrier—tumor 1		CRC_058-5	CRC	71	NT	NT	
		F_058	Maternal uncle		ID_058-4	M		*MSH2*	Exon 1-6 duplication	Exon duplication	Obligate carrier—tumor 2		Prostrate_058-5	Prostate	74	NT	NT	
		F_058	Maternal cousin	**13**	ID_058-5	F		*MSH2*	Exon 1-6 duplication	Exon duplication	Carrier		NA	Unaffected	Unknown	NT	NT	
	7	F_149	Index	**14**	ID_149 ^4^	M	8	*MSH2*	Exon 14-15 duplication	Exon duplication	Proband	8	CRC_149	CRC	28	MSH2/MSH6	NT	7
		F_149	Mother	**15**	ID_149-2 ^5^	F		*MSH2*	Exon 14-15 duplication	Exon duplication	Carrier		CRC_149-2	CRC	50	NT	NT	
	8	F_138	Index	**16**	ID_138	M	9	*MSH2*	c.328A > C p.(Lys110Gln)	Missense	Proband—tumor 1	9	CRC1_138	CRC	66	MSH2/MSH6	NT	8
		F_138	Index		ID_138	M		*MSH2*	c.328A > C p.(Lys110Gln)	Missense	Proband—tumor 2		CRC2_138	CRC	78	NT	NT	
		F_138	Brother	**17**	ID_138-2	M		*MSH2*	c.328A > C p.(Lys110Gln)	Missense	Carrier—tumor 1		Skin_138-2	Skin (melanoma)	43	NT	NT	
		F_138	Brother		ID_138-2	M		*MSH2*	c.328A > C p.(Lys110Gln)	Missense	Carrier—tumor 2		CRC_138-2	CRC	64	Normal	NT	
		F_138	Brother		ID_138-2	M		*MSH2*	c.328A > C p.(Lys110Gln)	Missense	Carrier—tumor 3		Pan_138-2	Pancreatic	76	Normal	NT	
	9	F_315	Index	**18**	ID_315	F	10	*MSH2*	c.1862G > T p.(Arg621Leu)	Missense	Proband—tumor 1	10	EC_315	EC	62	MSH2/MSH6	NT	9
		F_315	Index		ID_315	F		*MSH2*	c.1862G > T p.(Arg621Leu)	Missense	Proband—tumor 2		CRC_315	CRC	36	NT	NT	
		F_315	Sister	**19**	ID_315-2	F		*MSH2*	c.1862G > T p.(Arg621Leu)	Missense	Carrier—tumor	11	EC_315-2	EC	57	MLH1/PMS2 ^+^	POSITIVE	10
		F_315	Sister		ID_315-2	F		*MSH2*	c.1862G > T p.(Arg621Leu)	Missense	Carrier—polyp		Polyp_315-2	Benign endometrial polyp	59	NT	NT	
	10	F_008	Index	**20**	ID_008	F	11	*MSH2*	c.2005 + 3_2005 + 14del	Splice	Proband—tumor 1	12	EC_008	EC	62	MSH2/MSH6	NT	11
		F_008	Index		ID_008	F		*MSH2*	c.2005 + 3_2005 + 14del	Splice	Proband—tumor 2		Ureter_008	Ureter	45	NT	NT	
		F_008	Index		ID_008	F		*MSH2*	c.2005 + 3_2005 + 14del	Splice	Proband—tumor 3		CRC_008	CRC	51	NT	NT	
	11	F_132	Index	**21**	ID_132	F	12	*MSH2*	c.2060T > C p.(Leu687Pro)	Missense	Proband	13	CRC_132	CRC	37	MSH2/MSH6	NT	12
	12	F_156	Index	**22**	ID_156	F	13	*MSH6*	c.3556 + 5_3556 + 8delins p.?	Splice	Proband—tumor 1	14	CRC_156	CRC	61	MSH6	NT	13
		F_156	Index		ID_156	F		*MSH6*	c.3556 + 5_3556 + 8delins p.?	Splice	Proband—tumor 2	15	EC_156	EC	53	MSH6	NT	
		F_156	Brother	**23**	ID_156-2	M		*MSH6*	c.3556 + 5_3556 + 8delins p.?	Splice	Carrier		NA	Unaffected	Unknown	NT	NT	
	13	F_143	Index	**24**	ID_143	F	14	*PMS2*	c.137G > T p.(Ser46Ile)	Missense	Proband	16	EC_143	EC	29	PMS2	NEGATIVE	14
		F_143	Brother	**25**	ID_143-2	M		*PMS2*	c.137G > T p.(Ser46Ile)	Missense	Carrier		NA	Unaffected	Unknown	NT	NT	
		F_143	Father	**26**	ID_143-3	M		*PMS2*	c.137G > T p.(Ser46Ile)	Missense	Carrier		NA	Unaffected	Unknown	NT	NT	
		F_143	Paternal aunt	**27**	ID_143-4	F		*PMS2*	c.137G > T p.(Ser46Ile)	Missense	Carrier		NA	Unaffected	Unknown	NT	NT	
		F_143	Paternal aunt	**28**	ID_143-5	F		*PMS2*	c.137G > T p.(Ser46Ile)	Missense	Carrier		NA	Unaffected	Unknown	NT	NT	
		F_143	Paternal aunt	**29**	ID_143-6	F		*PMS2*	c.137G > T p.(Ser46Ile)	Missense	Carrier		NA	Unaffected	Unknown	NT	NT	
		F_143	Paternal grandmother	**30**	ID_143-7	F		*PMS2*	c.137G > T p.(Ser46Ile)	Missense	Obligate Carrier		NA	Unaffected	Unknown	NT	NT	
		F_143	Paternal aunt	**31**	ID_143-8	F		*PMS2*	c.137G > T p.(Ser46Ile)	Missense	Carrier		NA	Unaffected	Unknown	NT	NT	
PROBANDS’ MMR VUS DISCORDANT TO IHC PATTERN OF LOSS	14	F_328	Index	**32**	ID_328	F	15	*MLH1*	c.1153C > T p.(Arg385Cys)	Missense	Proband—tumor 1	17	CRC_328	CRC	41	MSH2/MSH6	NT	15
		F_328	Index		ID_328	F		*MLH1*	c.1153C > T p.(Arg385Cys)	Missense	Proband—tumor 2		Breast_328	Breast	54	NT	NT	
	15	F_202	Index	**33**	ID_202	F	16	*MSH2*	c.138C > G p.(His46Gln)	Missense	Proband—tumor 1	18	EC_202	EC	65	MSH6	NT	16
		F_202	Index		ID_202	F		*MSH2*	c.138C > G p.(His46Gln)	Missense	Proband—tumor 2		Breast_202	Breast	62	NT	NT	
	16	F_395	Index	**34**	ID_395	F	17	*MSH2*	c.668T > C p.(Leu223Pro)	Missense	Proband	19	CRC_395	CRC	38	MLH1/PMS2	NEGATIVE	17
	17	F_089	Index	**35**	ID_089	F	18	*MSH6*	c.2827G > T p.(Asp943Tyr)	Missense	Proband—tumor 1	20	CRC1_089	CRC	42	MLH1/PMS2	NEGATIVE	18
		F_089	Index		ID_089	F		*MSH6*	c.2827G > T p.(Asp943Tyr)	Missense	Proband—tumor 2		CRC2_089	CRC	44	NT	NT	
	18	F_046	Index	**36**	ID_046	F	19	*MSH6*	c.2950A > C p.(Asn984His)	Missense	Proband	21	EC_046	EC	70	PMS2	NEGATIVE	19
	19	F_263	Index	**37**	ID_263	F	20	*MSH6*	c.*85T > A p.?	3’ UTR	Proband	22	EC_263	EC	62	MLH1 ^+^/PMS2	NEGATIVE	20
	20	F_193	Index	**38**	ID_193	M	21	*PMS2*	c.2149G > A p.(Val717Met)	Missense	Proband	23	CRC_193	CRC	32	MLH1/PMS2	NEGATIVE	21
NO MMR IHC LOSS	21	F_170	Index	**39**	ID_170	F	22	*MLH1*	c.-117G > T p.?	5’ UTR	Proband	24	CRC_170	CRC	35	Normal	NEGATIVE	22
	22	F_111	Index	**40**	ID_111	F	23	*MSH6*	c.1153_1155del p.(Arg385del)	Inframe deletion	Proband	25	CRC_111	CRC	36	Normal	NT	23
	23	F_240	Index	**41**	ID_240	F	24	*PMS2*	c.2335G > A p.(Gly779Arg)	Missense	Proband	26	CRC_240	CRC	29	Normal	NEGATIVE	24
	24	F_352	Index	**42**	ID_352	M	25	*PMS2*	c.241G > A p.(Glu81Lys)	Missense	Proband—tumor 1	27	CRC1_352	CRC	58	Normal	NT	25
		F_352	Index		ID_352	M		*PMS2*	c.241G > A p.(Glu81Lys)	Missense	Proband—tumor 2	28	CRC2_352	CRC	58	MSH2/MSH6	NT	
		F_352	Index		ID_352	M		*PMS2*	c.241G > A p.(Glu81Lys)	Missense	Proband—tumor 3		Brain_352	Glioblastoma	52	NT	NT	

Abbreviations: ID, identification number; F, female; M, male; VUS, variant of uncertain clinical significance; CRC, colorectal cancer; EC, endometrial cancer; IHC, immunohistochemistry; NA, not applicable; NT, not tested; UTR, untranslated region. ^+^ Indicates heterogeneous/patchy loss of DNA mismatch repair protein expression by IHC. * The symbol indicates the variant is located in the 3’ untranslated region in the case of *MSH6* c.*85T>A and indicates the variant changes the amino acid to a stop codon in the case of *MSH6* c.4068_4071dup p.(Lys1358Aspfs*2). ^1^ Family members (F_161) were only tested for the germline *MLH1* c.539_541del p.(Val180del) variant and not for the concomitant *MSH6* c.4068_4071dup p.(Lys1358Aspfs*2) variant. ^2^ The parents (ID_376-2 and ID_376-3) from carrier ID_376 were tested for the germline *MSH2* exon 1-6 duplication and were found to be wildtype. ^3^ The sister (ID_058-3) and maternal cousin (ID_058-7) from carrier ID_058 were tested for the germline *MSH2* exon 1-6 duplication and were found to be wildtype. ^4^ Participant ID_149 carries a germline *MSH2* exon 14-15 duplication and a concomitant germline *POLE* c.1708C > A p.(Leu570Met) variant. ^5^ The mother (ID_149-2) of carrier ID_149 was tested for the familial germline *MSH2* exon 14-15 duplication and *POLE* c.1708C > A p.(Leu570Met) variants, however, was found to only carry the *MSH2* germline VUS.

**Table 2 cancers-15-04925-t002:** Display of the data used to implement the ACMG/InSiGHT framework for final DNA mismatch repair variant of uncertain significance classification.

							Incorporation of Tumor Data from This Study								**Incorporation of Externally Published Tumor Data**					
	#	Gene	Variant	Variant Type	ClinVar Variant ID	gnomAD (<1 in 50,000 Alleles)	Age at Diagnosis	MSI/MMR Status	Tumor *BRAF* V600E Mutation Status	Tumor *MLH1* Methylation Status	IHC	Total No. of Tumors with IHC Concordant to Germline VUS	Total No. of Tumors with IHC Discordant to Germline VUS	Total No. of Families with Germline MMR VUS Tested	**Prior Probability for Pathogenicity (MAPP/PP2 Score)**	**Tumor Characteristics Odds of Pathogenicity**	**Functional Data**	**Segregation Odds Ratio**	**SpliceAI Delta Score**	**ACMG/InSiGHT Classification**
MMR VUS CONCORDANT TO IHC PATTERN OF LOSS	1	*MLH1*	c.71_85del p.(Val24_Pro28del)	Inframe deletion	NA	0	59	MSI-H/dMMR (6/6)	WT	Negative	MLH1/PMS2	1	0	1	NA	0.41	None	NA	0.02	Class 3: VUS
	2	*MLH1*	c.539_541del p.(Val180del)	Inframe deletion	185186	0	59	MSI-H/dMMR (6/6)	WT	Negative	MLH1/PMS2	1	0	1	NA	0.41	None	NA	0.01	Class 3: VUS
	3	*MLH1*	c.1594G > C p.(Gly532Arg)	Missense	NA	0	47, 54	MSI-H/dMMR (5/6) (1×), MSI-H/dMMR (6/6) (1×)	WT (2×)	Negative (2×)	MLH1 ^+^/PMS2 (1×), MLH1/PMS2 (1×)	2	0	1	0.96	6.52	None	NA	0.01	Class 3: VUS
	4	*MLH1*	c.1595G > A p.(Gly532Asp)	Missense	976474	0	41	MSI-H/dMMR (5/6)	WT	Negative	MLH1/PMS2	3	0	5	0.96	153.04	None	1.96	0.01	Class 4: LP
	5&6	*MSH2*	Duplication exon 1–6 ^1^	Exon duplication	NA	0	29, 49	MSI-H/dMMR (6/6) (2×)	WT (2×)	NT	MSH2/MSH6 (2×)	2	0	2	NA	42.51	None	NA	NA	Class 3: VUS
	7	*MSH2*	Duplication exon 14–15	Exon duplication	NA	0	28	MSI-H/dMMR (6/6)	WT	NT	MSH2/MSH6	1	0	1	NA	6.52	None	NA	NA	Class 4: LP
	8	*MSH2*	c.328A > C p.(Lys110Gln)	Missense	127642	0.00000398	66	MSI-H/dMMR (6/6)	WT	NT	MSH2/MSH6	1	0	1	0.06	4.06	None	NA	0.01	Class 3: VUS
	9	*MSH2*	c.1862G > T p.(Arg621Leu)	Missense	218040	0.0000319	65	MSI-H/dMMR (6/6)	WT	NT	MSH2/MSH6	5	0	5	0.85	1103.14	None	1.32	0	Class 4: LP
	10	*MSH2*	c.2005 + 3_2005 + 14del	Splice	90842	0	51	MSI-H/dMMR (6/6)	WT	NT	MSH2/MSH6	2	0	2	0.26	16.48	None	1.44	0.57	Class 4: LP
	11	*MSH2*	c.2060T > C p.(Leu687Pro)	Missense	90873	0	37	MSI-H/dMMR (6/6)	WT	NT	MSH2/MSH6	1	0	>1 ^2^	0.96	6.52	None	NA	0	Class 4: LP
	12	*MSH6*	c.3556 + 5_3556 + 8delins	Splice	NA	0	61	MSI-H/dMMR (5/6)	WT	NT	MSH6	3	0	3	0.26	66.92	None	NA	0	Class 4: LP
	13	*PMS2*	c.137G > T p.(Ser46Ile)	Missense	9245	0.000163	29	MSI-H/dMMR (6/6)	WT	Negative	PMS2	2	0	2	0.939	0.65	None	3.083	0.03	Class 4: LP
MMR VUS DISCONCORDANT TO IHC PATTERN OF LOSS	14	*MLH1*	c.1153C > T p.(Arg385Cys)	Missense	89653	0.0000597	41	MSI-H/dMMR (6/6)	WT	NT	MSH2/MSH6	0	1	2	0.937	0.01	None	9.45	0.03	Class 3: VUS
	15	*MSH2*	c.138C > G p.(His46Gln)	Missense	90654	0.000218	65	MSI-H/dMMR (3/6)	WT	NT	MSH6	1	3	4	0.74	0.00014	None	0.14	0	Class 2: LB
	16	*MSH2*	c.668T > C p.(Leu223Pro)	Missense	408456	0	38	MSI-H/dMMR (6/6)	WT	Negative	MLH1/PMS2	0	1	1	0.525	0.11	None	NA	0	Class 3: VUS
	17	*MSH6*	c.2827G > T p.(Asp943Tyr)	Missense	142495	0.000046	42	MSI-H/dMMR (6/6)	WT	Negative	MLH1/PMS2	0	1	1	0.05	0.11	None	NA	0	Class 3: VUS
	18	*MSH6*	c.2950A > C p.(Asn984His)	Missense	186492	0.0000479	70	MSI-H/dMMR (4/6)	WT	Negative	PMS2	0	1	1	0.005	0.11	None	NA	0	Class 3: VUS
	19	*MSH6*	c.4068_4071dup p.(Lys1358Aspfs*2)	Frameshift	89518		59	MSI-H/dMMR (6/6)	WT	-	MLH1/PMS2	1	0	1	NA	4.06	None	NA	0.1	Class 2: LB
	20	*MSH6*	c.*85T > A	3’ UTR	89155	0.00786	62	MSI-H/dMMR (5/6)	WT	Negative	MLH1 ^+^/PMS2	0	4	4	0	0	None	NA	0	Class 1: B
	21	*PMS2*	c.2149G > A p.(Val717Met)	Missense	41709	0.000758	32	MSI-H/dMMR (5/6)	WT	Negative	MLH1/PMS2	2	0	2	0.595	26.47	None	0.308	0	Class 3: VUS
NO MMR IHC LOSS	22	*MLH1*	c.-117G > T	5’ UTR	344901	0	35	MSI-H/dMMR (3/6)	WT	NT	Normal	0	1	1	NA	0.11	None	NA	0.07	Class 3: VUS
	23	*MSH6*	c.1153_1155del p.(Arg385del)	Inframe deletion	89177	0	36	MSI-H/dMMR (5/6)	WT	NT	Normal	1	1	4	0.5	0.46	None	15.22	0	Class 3: VUS
	24	*PMS2*	c.2335G > A p.(Gly779Arg)	Missense	127778	0.00000799	29	MSS/pMMR (1/6)	WT	Negative	Normal	1	0	1	0.96	0.11	None	NA	0.03	Class 3: VUS
	25	*PMS2*	c.241G > A p.(Glu81Lys)	Missense	182817	0.0000159	58	MSI-H/dMMR (5/6) (1×), MSS/pMMR (1/6) (1×)	WT (2x)	NT	MSH2/MSH6 (1×), Normal (1×)	0	2	2	0.045	0.02	None	NA	0	Class 3: VUS

Abbreviations: MSI, microsatellite instability; MSS, microsatellite stable; MMR, DNA mismatch repair; dMMR, DNA mismatch repair deficiency; pMMR, DNA mismatch repair proficiency; WT, wildtype; IHC, immunohistochemistry; MAPP, multi-variate analysis of protein polymorphisms; PP2, PolyPhen-2.1; VUS, variant of uncertain clinical significance; LB, likely benign; LP, likely pathogenic; NA, not applicable; NT, not tested. ^+^ Indicates heterogeneous/patchy loss of DNA mismatch repair protein expression by IHC. * The symbol indicates the variant is located in the 3’ untranslated region in the case of *MSH6* c.*85T > A and indicates the variant changes the amino acid to a stop codon in the case of *MSH6* c.4068_4071dup p.(Lys1358Aspfs*2). ^1^ Two unrelated families (F_058 and F_376) who carry the *MSH2* exon 1–6 duplication were grouped together for the purposes of the ACMG/InSiGHT classification; however, at this stage without further investigation, it remains unknown if the breaking points of the variants identified in each family are in the same location. ^2^ Multiple entries documented in Ambry Genetics (https://www.ambrygen.com/, last accessed date: 23 May 2023).

**Table 3 cancers-15-04925-t003:** Overview of targeted tumor sequencing results from the test group including sequenced family members.

			Germline MMR VUS				Tumor Molecular Data				Lynch Syndrome Associated Tissue Features			
#	Carrier ID	Tissue ID	Gene	Base Change	Protein Change	ACMG/InSiGHT Classification	Tissue	IHC	*MLH1* Methylation	Tumor Mutation Burden (Mutations/Megabase)	MSI/MMR Status by Additive Feature Approach	Presence of a Somatic Second Hit in MMR Gene Harboring VUS	Presence of a Somatic Mutation NOT in MMR Gene Harboring VUS	Final Tumour Classification
1	ID_326	CRC_326	*MLH1*	c.1595G > A	p.(Gly532Asp)	Class 4: LP	CRC	MLH1 ^+^/PMS2	NEGATIVE	110.7	MSI-H/dMMR (5/6)	1× LOH (*MLH1*)	None	dMMR—VUS + 2nd hit (*MLH1*)
2	ID_149	CRC_149	*MSH2*	Exon 14-15 duplication	p.?	Class 4: LP	CRC	MSH2/MSH6	NT	108.2	MSI-H/dMMR (6/6)	1× mut. (*MSH2*)	None	dMMR—VUS + 2nd hit (*MSH2*)
3	ID_315	EC_315	*MSH2*	c.1862G > T	p.(Arg621Leu)	Class 4: LP	EC	MSH2/MSH6	NT	77.3	MSI-H/dMMR (6/6)	1× mut. (*MSH2*)	1× mut. (*MSH6*) and 1× mut. (*PMS2*)	dMMR—VUS + 2nd hit (MSH2)
4	ID_315-2	EC_315-2	*MSH2*	c.1862G > T	p.(Arg621Leu)	Class 4: LP	EC	MLH1/PMS2 ^+^	*POSITIVE*	55.4	MSS/pMMR (0/6)	None	None	dMMR—*MLH1* methylation
5	ID_008	EC_008	*MSH2*	c.2005 + 3_2005 + 14del	p.?	Class 4: LP	EC	MSH2/MSH6	NT	140.6	MSI-H/dMMR (6/6)	1× LOH (*MSH2*)	1× mut. (*MLH1*) and 1× mut. (*PMS2*)	dMMR—VUS + 2nd hit (*MSH2*)
6	ID_132	CRC_132	*MSH2*	c.2060T > C	p.(Leu687Pro)	Class 4: LP	CRC	MSH2/MSH6	NT	115.2	MSI-H/dMMR (6/6)	1× mut. (*MSH2*)	None	dMMR—VUS + 2nd hit (*MSH2*)
7	ID_156	CRC_156	*MSH6*	c.3556 + 5_3556 + 8delins	p.?	Class 4: LP	CRC	MSH6	NT	79.8	MSI-H/dMMR (5/6)	1× mut. (*MSH6*)	None	dMMR—VUS + 2nd hit (*MSH6*)
8		EC_156				Class 4: LP	EC	MSH6	NT	195	MSI-H/dMMR (3/6)	None	None	dMMR—VUS (*MSH6*)
9	ID_143	EC_143	*PMS2*	c.137G > T	p.(Ser46Ile)	Class 4: LP	EC	PMS2	NEGATIVE	649.9	MSI-H/dMMR (6/6)	2× mut. (*PMS2*)	2× mut. (*MSH2*), 2× mut. (*MSH6*) and 1× mut. (*MLH1*)	dMMR—VUS + 2nd hit (*PMS2*)
10	ID_161 ^1^	EC_161	*MLH1*	c.539_541del	p.(Val180del)	Class 3: VUS	EC	MLH1/PMS2	NEGATIVE	99.8	MSI-H/dMMR (6/6)	1× mut. (*MLH1*)	None	dMMR—VUS + 2nd hit (*MLH1*)
11	ID_051	EC_051	*MLH1*	c.71_85del	p.(Val24_Pro28del)	Class 3: VUS	EC	MLH1/PMS2	NEGATIVE	418	MSI-H/dMMR (6/6)	1× LOH (*MLH1*) and 1x mut. (*MLH1*)	1× mut. (*MSH6*)	dMMR—VUS + 2nd hit (*MLH1*)
12	ID_176	CRC_176	*MLH1*	c.1594G > C	p.(Gly532Arg)	Class 3: VUS	CRC	MLH1/PMS2	NEGATIVE	102.7	MSI-H/dMMR (6/6)	1× LOH (*MLH1*)	1× mut. (*MSH6*)	dMMR—VUS + 2nd hit (*MLH1*)
13		EC_176				Class 3: VUS	EC	MLH1 ^+^/PMS2	NEGATIVE	52.4	MSI-H/dMMR (5/6)	2× mut. (*MLH1*)	1× mut. (*MSH6*)	dMMR—VUS + 2nd hit (*MLH1*)
14	ID_058	EC_058	*MSH2*	Exon 1-6 duplication	p.?	Class 3: VUS	EC	MSH2/MSH6	NT	102.2	MSI-H/dMMR (6/6)	1× LOH (*MSH2*)	None	dMMR—VUS + 2nd hit (*MSH2*)
15	ID_138	CRC_138	*MSH2*	c.328A > C	p.(Lys110Gln)	Class 3: VUS	CRC	MSH2/MSH6	NT	189	MSI-H/dMMR (6/6)	2× mut. (*MSH2*)	2× mut. (*MSH6*)	dMMR—VUS + 2nd hit (*MSH2*)
16	ID_170	CRC_170	*MLH1*	c.-117G > T	p.?	Class 3: VUS	CRC	Normal	NEGATIVE	10	MSI-H/dMMR (3/6)	1× LOH (*MLH1*)	None	dMMR—VUS + 2nd hit (*MLH1*)
17	ID_111	CRC_111	*MSH6*	c.1153_1155del	p.(Arg385del)	Class 3: VUS	CRC	Normal	NT	40.4	MSI-H/dMMR (5/6)	1× LOH (*MSH6*)	None	dMMR—VUS + 2nd hit (*MSH6*)
18	ID_376	CRC_376	*MSH2*	Exon 1-6 duplication	p.?	Class 3: VUS	CRC	MSH2/MSH6	NT	156.6	MSI-H/dMMR (6/6)	None	1× mut. (*MSH6*) and 1× mut. (*PMS2*)	dMMR—VUS (*MSH2*)
19	ID_328	CRC_328	*MLH1*	c.1153C > T	p.(Arg385Cys)	Class 3: VUS	CRC	MSH2/MSH6	NT	39.9	MSI-H/dMMR (6/6)	None	1× LOH (*MSH2*) and 1× mut. (*MSH2*)	dMMR—double somatic (*MSH2*)
20	ID_395	CRC_395	*MSH2*	c.668T > C	p.(Leu223Pro)	Class 3: VUS	CRC	MLH1/PMS2	NEGATIVE	111.2	MSI-H/dMMR (6/6)	None	1× LOH (*MLH1*) and 1× mut. (*MLH1*)	dMMR—double somatic (*MLH1*)
21	ID_089	CRC_089	*MSH6*	c.2827G > T	p.(Asp943Tyr)	Class 3: VUS	CRC	MLH1/PMS2	NEGATIVE	143.6	MSI-H/dMMR (6/6)	None	1× LOH (*MLH1*) and 2× mut. (*MLH1*)	dMMR—double somatic (*MLH1*)
22	ID_046	EC_046	*MSH6*	c.2950A > C	p.(Asn984His)	Class 3: VUS	EC	PMS2	NEGATIVE	371.1	MSI-H/dMMR (4/6)	1× mut. (*MSH6*)	2× mut. (*PMS2*)	dMMR—double somatic (*PMS2*)
23	ID_352 ^2^	CRC2_352	*PMS2*	c.241G > A	p.(Glu81Lys)	Class 3: VUS	CRC	MSH2/MSH6	NT	162.6	MSI-H/dMMR (5/6)	None	2× mut. (*MSH2*), 2× mut. (*MLH1*) and 1× mut. (*MSH6*)	dMMR—double somatic (*MSH2*)
24	ID_193	CRC_193	*PMS2*	c.2149G > A	p.(Val717Met)	Class 3: VUS	CRC	MLH1/PMS2	NEGATIVE	79.3	MSI-H/dMMR (5/6)	None	2× mut. (*MLH1*)	dMMR—double somatic (*MLH1*)
25	ID_263	EC_263	*MSH6*	c.*85T > A	p.?	Class 1: B	EC	MLH1 ^+^/PMS2	NEGATIVE	50.4	MSI-H/dMMR (5/6)	1× mut. (*MSH6*)	1× LOH (*MLH1*) and 1× mut. (*MLH1*)	dMMR—double somatic (*MLH1*)
26	ID_202	EC_202	*MSH2*	c.138C > G	p.(His46Gln)	Class 2: LB	EC	MSH6	NT	482.8	MSI-H/dMMR (3/6)	1× mut. (*MSH2*)	2× mut. (*MSH6*)	dMMR—double somatic (*MSH6*)
27	ID_240	CRC_240	*PMS2*	c.2335G > A	p.(Gly779Arg)	Class 3: VUS	CRC	Normal	NEGATIVE	17.5	MSS/pMMR (0/6)	None	None	pMMR
28	ID_352 ^2^	CRC1_352	*PMS2*	c.241G > A	p.(Glu81Lys)	Class 3: VUS	CRC	Normal	NT	208	MSS/pMMR (1/6)	None	2× mut. (*MLH1*) and 2× mut. (*MSH6*)	pMMR
29	ID_161 ^1^	EC_161	*MSH6*	c.4068_4071dup	p.(Lys1358Aspfs*2)	Class 2: LB	EC	MLH1/PMS2	NEGATIVE	99.8	MSI-H/dMMR (6/6)	None	1× mut. (*MLH1*)	VUS benign

Abbreviations: ID, identification number; CRC, colorectal cancer; EC, endometrial cancer; LP, likely pathogenic; VUS, variant of uncertain clinical significance; LB, likely benign; B, benign; IHC, immunohistochemistry; MMR, DNA mismatch repair; MSI, microsatellite instability; MSI-H, high levels of microsatellite stability; MSS, microsatellite stable; dMMR, DNA mismatch repair deficient; pMMR, DNA mismatch repair proficient; LOH, loss of heterozygosity; mut., single somatic mutation; NT, not tested. ^+^ Indicates heterogeneous/patchy loss of DNA mismatch repair protein expression by IHC. * The symbol indicates the variant is located in the 3’ untranslated region in the case of *MSH6* c.*85T > A and indicates the variant changes the amino acid to a stop codon in the case of *MSH6* c.4068_4071dup p.(Lys1358Aspfs*2). ^1^ Participant (ID_161) developed a single endometrial cancer showing loss of MLH1/PMS2 by immunohistochemistry but carried two VUS; one in *MLH1* and one in *MSH6*. ^2^ Participant (ID_352) carried a *PMS2* VUS but developed two different CRCs; one with loss of MSH2/MSH6 protein expression and one with no loss of MMR protein expression (pMMR).

**Table 4 cancers-15-04925-t004:** Overview of normal tissue screening for DNA mismatch repair deficient crypts and glands in the reference and test groups.

									Evaluation of Screening Assays for Potential Addition to Current Variant Classification Approaches				
#	Carrier Type	Carrier ID	Tissue	IHC	*MLH1* Methylation	Gene	Variant	ACMG/InSiGHT Classification	MSI-H/dMMR by Additive Feature Approach	Presence of a Somatic Second Hit	Presence of a dMMR Crypt/Gland	Amount Screened	Final Tumor Classification
1	Reference	Ref_029	CRC	MLH1/PMS2	NT	*MLH1*	c.1713_1716delTGGT p.Phe571Leufs*19	Class 5: P	Yes	Yes	Yes	2 × 80 µM	dMMR—LS + 2nd hit (*MLH1*)
2	Reference	Ref_029-2 ^1^	CRC	MLH1/PMS2	NT	*MLH1*	c.1713_1716delTGGT p.Phe571Leufs*19	Class 5: P	NA	NA	Yes	10 × 4 µM	dMMR—LS (*MLH1*)
3	Reference	Ref_605	CRC	MLH1/PMS2	NT	*MLH1*	c.1852_1854delAAG p.Lys618del	Class 5: P	Yes	Yes	Yes	1 × 80 µM	dMMR—LS + 2nd hit (*MLH1*)
4	Reference	Ref_411	CRC	MSH2/MSH6	NT	*MSH2*	c.1889_1892delGAAG p.Gly630Glufs*4	Class 5: P	Yes	Yes	No ^2^	2 × 80 µM	dMMR—LS + 2nd hit (*MSH2*)
5	Reference	Ref_897 ^3^	CRC	Normal	NT	-	Wildtype	NA	NA	NA	No	3 × 80 µM	pMMR—non-LS
6	Reference	Ref_972 ^3^	CRC	Normal	NT	-	Wildtype	NA	NA	NA	No	3 × 80 µM	pMMR—non-LS
7	Test—Proband	ID_051	EC	MLH1/PMS2	NEGATIVE	*MLH1*	c.71_85del p.(Val24_Pro28del)	Class 3: VUS	Yes	Yes	No	10 × 4 µM	dMMR—VUS + 2nd hit (*MLH1*)
8	Test—Proband	ID_161	EC	MLH1/PMS2	NEGATIVE	*MLH1*	c.539_541del p.(Val180del)	Class 3: VUS	Yes	Yes	No	3 × 80 µM	dMMR—VUS + 2nd hit (*MLH1*)
9	Test—Proband	ID_176	CRC	MLH1/PMS2	NEGATIVE	*MLH1*	c.1594G > C p.(Gly532Arg)	Class 3: VUS	Yes	Yes	No	3 × 80 µM	dMMR—VUS + 2nd hit (*MLH1*)
10	Test—Proband	ID_326	CRC	MLH1 ^+^/PMS2	NEGATIVE	*MLH1*	c.1595G > A p.(Gly532Asp)	Class 4: LP	Yes	Yes	No ^2^	2 × 80 µM	dMMR—VUS + 2nd hit (*MLH1*)
11	Test—Proband	ID_058	EC	MSH2/MSH6	NT	*MSH2*	Exon 1-6 duplication	Class 3: VUS	Yes	Yes	Yes	1 × 80 µM	dMMR—VUS + 2nd hit (*MSH2*)
12	Test—Proband	ID_376	CRC	MSH2/MSH6	NT	*MSH2*	Exon 1-6 duplication	Class 3: VUS	Yes	No	No	3 × 80 µM	dMMR—VUS (*MSH2*)
13	Test—Proband	ID_138	CRC	MSH2/MSH6	NT	*MSH2*	c.328A > C p.(Lys110Gln)	Class 3: VUS	Yes	Yes	No	1 × 4 µM	dMMR—VUS + 2nd hit (*MSH2*)
14	Test—Relative	ID_315-2	EC	MLH1/PMS2 ^+^	POSITIVE	*MSH2*	c.1862G > T p.(Arg621Leu)	Class 4: LP	No	No	Yes	1 × 80 µM	dMMR—*MLH1* methylation
15	Test—Proband	ID_132	CRC	MSH2/MSH6	NT	*MSH2*	c.2060T > C p.(Leu687Pro)	Class 4: LP	Yes	Yes	No ^4^	10 × 4 µM	dMMR—VUS + 2nd hit (*MSH2*)
16	Test—Proband	ID_156	CRC	MSH6	NT	*MSH6*	c.3556 + 5_3556 + 8delins	Class 4: LP	Yes	Yes	Yes	10 × 4 µM	dMMR—VUS + 2nd hit (*MSH2*)
17	Test—Proband	ID_046	EC	PMS2	NEGATIVE	*MSH6*	c.2950A > C p.(Asn984His)	Class 3: VUS	Yes	Yes	No	3 × 80 µM	dMMR—double somatic (*PMS2*)
18	Test—Proband	ID_143	EC	PMS2	NEGATIVE	*PMS2*	c.137G > T p.(Ser46Ile)	Class 4: LP	Yes	Yes	Failed test	3 × 80 µM	dMMR—VUS + 2nd hit (*PMS2*)

Abbreviations: ID, identification number; CRC, colorectal cancer; EC, endometrial cancer; IHC, immunohistochemistry; P, pathogenic; LP, likely pathogenic; VUS, variant of uncertain significance; LB, likely benign; MMR, DNA mismatch repair; dMMR, DNA mismatch repair deficiency; pMMR, DNA mismatch repair proficiency; MSI-H, high levels of microsatellite stability; MSS, microsatellite stable; NA, not applicable; NT, not tested; LS, Lynch syndrome; MLH1me, *MLH1* gene promoter hypermethylation. ^+^ Indicates heterogeneous/patchy loss of DNA mismatch repair protein expression by IHC. ^1^ Maternal aunt of Ref_029, who is also carrier of the family *MLH1* pathogenic variant, was identified to have a dMMR crypt in normal colonic mucosa, but their CRC did not undergo tumor sequencing. ^2^ Block was depleted after screening of 2 × 80 µM of normal tissue. ^3^ This sample did not undergo next-generation sequencing. ^4^ Only received slides.

## Data Availability

The original contributions presented in the study are included in the article/Appendix A, further inquiries can be directed to the corresponding author.

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
