# Peer review of "DNA Mismatch Repair Gene Variant Classification: Evaluating the Utility of Somatic Mutations and Mismatch Repair Deficient Colonic Crypts and Endometrial Glands"

_cancers, 2023, doi:10.3390/cancers15204925_

Round 1

Reviewer 1 Report

Identification of a germline pathogenic variant in one of the MMR genes is necessary to establish a diagnosis of Lynch Syndrome both in subjects with a personal / family history of cancers fulfilling approved risk criteria and in patients with tumors showing loss of expression of one or more of the mismatch repair gene products by IHC, an MSI status demonstrated by direct MSI testing or NGS or an identified pathogenic MMR gene mutation.

Over the last few decades, there has been great insight into the pathogenesis and molecular diagnosis of Lynch Syndrome with thousands of unique germline MMR genes variants identified and recorded in international databases. However, relevant portions of the identified MMR genes variants do not have a proven pathogenic role (variants of uncertain clinical significance, VUS) leaving uncertainties as to syndrome diagnosis and therefore optimal cancer surveillance.

In their work, Walker et al. have analyzed a group of 25 patients with colororectal and/or endometrial cancer, hatrboring germline molecular alterations of the MMR genes of uncertain clinical significance (VUS) and reclassified these variants based on the assessment of the presence of somatic MMR co-mutations in the tumor tissue and of a defective MMR protein expression in normal colonic crypts or endometrial glands of surrounding normal tissue. This allowed to reclassify 10 of 25 VUS (7 as likely pathogenic and 3 as benign/likely benign). 

The methodology was appropriate and the results clearly described. The author’s conclusion that this study confirms that assessment of these somatic features of Lynch syndrome can improve the classification of germline MMR variants, potentially allowing improved clinical care, is supported by their findings. The discussion should be substantially shortened.

As tumor samples were analysed with NGS testing, it would be interesting to report also data on mutational tumor burden for the different variants given the potential value for treatment with ICI.

Author Response

Response to Reviewer 1 Comments

1. Summary

Thank you very much for taking the time to review this manuscript. Please find the detailed responses below and the corresponding revisions/corrections in track changes in the re-submitted files.

2. Questions for General Evaluation

Reviewer’s Evaluation

Response and Revisions

Does the introduction provide sufficient background and include all relevant references?

Yes

No response or revision required.

Are all the cited references relevant to the research?

Yes

Is the research design appropriate?

Yes

Are the methods adequately described?

Yes

Are the results clearly presented?

Yes

Are the conclusions supported by the results?

Yes

3. Point-by-point response to Comments and Suggestions for Authors

Comments 1: The discussion should be substantially shortened.

Response 1: We have removed text in the discussion as recommended by the Reviewer.

Comments 2: As tumor samples were analysed with NGS testing, it would be interesting to report also data on mutational tumor burden for the different variants given the potential value for treatment with ICI.

Response 2: We have added the tumor mutational burden for each variant to Table 3 and have added following text to the Materials and Methods section titled 2.5 Bioinformatics Pipeline.

Text added: “The tumor mutation burden was calculated as the number of somatic single nucleotide and INDEL mutations divided by the target region (megabase).”

4. Response to Comments on the Quality of English Language

Point 1: English language fine. No issues detected.

Response 1: Not applicable.

5. Other minor changes to the manuscript

1)     We have changed the section titled “Author Contributions” to follow the ICMJE guidelines, as requested by the editorial management;

2)     We have put all tables into the main text, as requested by the editorial management;

3)     We have changed the wording from “AgeDx” to “Age at diagnosis” in Table 2 and in the footnote to be consistent throughout the manuscript.

Reviewer 2 Report

This manuscript introduces and comprehensively evaluates Lynch syndrome-specific tumor features for their ability to support the ACMG/InSiGHT framework in classifying variants of uncertain clinical significance (VUS) in the MMR genes.

The topic is original and relevant to the field. There is limited information on this topic in the literature.

This article makes clear that identifying some specific Lynch syndrome features can improve MMR variant classification, enabling optimal clinical care.

There are no further improvements regarding the methodology.

The conclusions are consistent with the evidence and arguments presented as well as summarize the main point of this article. 

References are up-to-date and appropriate

Tables and figures are well formatted and make the study easy to follow

Minor revision

In the last few years, technological developments in the medical  field have been rapid and are continuously evolving. One of the most revolutionizing breakthroughs was the introduction of the IoT concept within medical practice.”

Add this information in the discussion section and explain the role of IoT in evaluating the utility of somatic mutations  Consider citing the article on the Internet of Surgical Things

https://pubmed.ncbi.nlm.nih.gov/35746359/

Author Response

Response to Reviewer 2 Comments

1. Summary

Thank you very much for taking the time to review this manuscript. Please find the detailed responses below and the corresponding revisions/corrections in track changes in the re-submitted files.

2. Questions for General Evaluation

Reviewer’s Evaluation

Response and Revisions

Does the introduction provide sufficient background and include all relevant references?

Can be improved

No revisions were made. We and Reviewer 1 were happy with the introduction.

Are all the cited references relevant to the research?

Yes

Is the research design appropriate?

Yes

Are the methods adequately described?

Yes

Are the results clearly presented?

Yes

Are the conclusions supported by the results?

Yes

3. Point-by-point response to Comments and Suggestions for Authors

Comments 1: “In the last few years, technological developments in the medical field have been rapid and are continuously evolving. One of the most revolutionizing breakthroughs was the introduction of the IoT concept within medical practice.”

Add this information in the discussion section and explain the role of IoT in evaluating the utility of somatic mutations. Consider citing the article on the Internet of Surgical Things

https://pubmed.ncbi.nlm.nih.gov/35746359/.

Response 1: Thank you for your comment. However, we do not see the relevance of citing this publication in our manuscript. Not one of our key terms, such as “somatic”, “mutations”, “next-generation sequencing”, “variant”, “classification”, “DNA mismatch repair deficiency”, “crypts / glands” or others were mentioned at least once in this article. Therefore we have not made any changes to the manuscript.

4. Response to Comments on the Quality of English Language

Point 1: English language fine. No issues detected.

Response 1: Not applicable.

5. Other minor changes to the manuscript

1)     We have changed the section titled “Author Contributions” to follow the ICMJE guidelines, as requested by the editorial management;

2)     We have put all tables into the main text, as requested by the editorial management;

3)     We have changed the wording from “AgeDx” to “Age at diagnosis” in Table 2 and in the footnote to be consistent throughout the manuscript.
